# Sediment release of dissolved organic matter to the oxygen minimum zone off Peru

Alexandra N. Loginova[1], Andrew W. Dale[1], Frédéric A. C. Le Moigne[1,2], Sören Thomsen[1,3], Stefan Sommer[1], David Clemens[1], Klaus Wallmann[1], and Anja Engel[1]

[1]GEOMAR Helmholtz Centre for Ocean Research Kiel, Germany
[2]Mediterranean Institute of Oceanography, UM110, Aix Marselle Université, CNRS, IRD, 13288, Marselle, France
[3]LOCEAN-IPSL, IRD/CNRS/Sorbonnes Universites (UPMC)/MNHN, Paris, France

**Correspondence:** Anja Engel (aengel@geomar.de)

**Abstract.**

The eastern tropical South Pacific (ETSP) represents one of the most productive areas in the ocean that is characterised by a pronounced oxygen minimum zone (OMZ). Particulate organic matter (POM) that sinks out of the euphotic zone is supplied to the anoxic sediments and utilised by microbial communities. The degradation of POM is associated with the production and reworking of dissolved organic matter (DOM). The release of DOM to the overlying waters may represent an important organic matter escape mechanism from remineralisation within sediments but received little attention in OMZ regions so far. Here, we combine measurements of dissolved organic carbon (DOC) and dissolved organic nitrogen (DON) with DOM optical properties in the form of chromophoric (CDOM) and fluorescent (FDOM) DOM from porewaters and near-bottom waters of the ETSP off Peru. We evaluate diffusion-driven fluxes and net *in situ* fluxes of DOC and DON to investigate processes affecting DOM cycling at the sediment-water interface along a transect at $12^{\circ}$S. To our knowledge, these are the first data for sediment release of DON and porewater CDOM and FDOM for the ETSP off Peru. Porewater DOC accumulated with increasing sediment depth, suggesting an imbalance between DOM production and remineralisation within sediments. High DON accumulation resulted in very low porewater DOC/DON ratios ($\leq 1$) which could be caused by an "uncoupling" in DOC and DON remineralisation. Diffusion driven fluxes of DOC and DON exhibited high spatial variability and ranged from $0.2\pm0.1$ $\mathrm{mmol\,m^{-2}d^{-1}}$ to $2.5\pm1.3$ $\mathrm{mmol\,m^{-2}d^{-1}}$ and from $-0.04\pm0.02$ $\mathrm{mmol\,m^{-2}d^{-1}}$ to $3.3\pm1.7$ $\mathrm{mmol\,m^{-2}d^{-1}}$, respectively. Generally low net *in situ* DOC and DON fluxes, as well as a steepening of spectral inclination (*S*) of CDOM and an increase in humic-like DOM at the sediment-water interface over time indicated active microbial DOM utilisation. The latter may potentially be stimulated by the presence of nitrate ($NO_3^-$) and nitrite ($NO_2^-$) in the water column. The microbial DOC utilisation rates, estimated in our study, are potentially sufficient to support denitrification rates of $0.2$-$1.4$ $\mathrm{mmol\,m^{-2}d^{-1}}$, suggesting that the sediment release of DOM may on occasion contribute to nitrogen loss processes in the ETSP off Peru.

## 1 Introduction

The eastern tropical South Pacific (ETSP) is one of the most productive areas of the world ocean (Pennington et al., 2006). High productivity, followed by intensive organic matter remineralisation (e.g. Loginova et al., 2019; Maßmig et al., 2020) in

combination with sluggish ventilation (Stramma et al., 2005; Keeling et al., 2010) leads to the formation of a pronounced oxygen minimum zone (OMZ) (e.g. Stramma et al., 2008). Remineralisation of organic matter under anoxia induces nitrogen (N)-loss by denitrification, anammox as well as dissimilatory nitrate reduction to ammonium (DNRA) in the water column and sediments off the coast of Peru (Kalvelage et al., 2013; Arévalo-Martínez et al., 2015; Dale et al., 2016; Sommer et al., 2016;

Glock et al., 2019). Although organic matter remineralisation is classically assumed to be limited by the absence of oxygen (Demaison and Moore, 1980), recent studies report similar abilities of marine microbes to degrade organic matter in oxygenated surface waters and within OMZs (Pantoja et al., 2009; Maßmig et al., 2019, 2020), suggesting that other factors, such as the quality of organic matter may regulate microbial activity within OMZs (Pantoja et al., 2009; Le Moigne et al., 2017). Similar to water column studies, extensive fieldwork campaigns conducted on sediments off Peru also suggested intensive particulate

organic matter (POM) remineralisation under full anoxia (Dale et al., 2015).

While POM degradation in sediments is mostly associated with its full remineralisation to dissolved inorganic carbon (DIC) and inorganic nutrients, the mechanism of POM remineralisation implies important intermediate stages of dissolved organic matter (DOM) production, reworking, and mineralisation (Smith et al., 1992; Komada et al., 2013). Thus, around 10 % of remineralised particulate organic carbon (POC) may accumulate as dissolved organic carbon (DOC) in the porewaters (Alperin

et al., 1999). In turn, DOM efflux may represent an important escape mechanism for carbon from sediments (e.g. Ludwig et al., 1996; Burdige et al., 1999) and a source of organic matter to the water column (e.g. Burdige et al., 2016). Despite the importance of sediment DOM for organic matter cycling, the measurements of benthic DOM fluxes remain scarce, and the reactivity of the porewater DOM is not well constrained.

The release of dissolved substances from anoxic sediments is regulated mainly by diffusion through the sediment-water

interface (e.g. Lavery et al., 2001, and references therein). Diffusion-driven DOM fluxes (hereafter named "diffusive fluxes") and net DOM fluxes (hereafter termed " *in situ* net fluxes") are commonly evaluated from porewater gradients using Fick's First Law and by enclosing and incubating a small area of the sediment surface over time, respectively. Diffusive DOM fluxes are, generally, consistent with net DOM fluxes in non-bioturbated anoxic sediments (Burdige et al., 1992). In some sediments, however, the diffusive flux may overestimate the net flux (Burdige et al., 1992; Lavery et al., 2001). This overestimation may

be attributed to bioturbation, "unfavourable" redox conditions (Lavery et al., 2001), irreversible adsorption onto particles, and biological DOM consumption at the sediment-water interface or in the bottom waters (Burdige et al., 1992). The determination of *in situ* net DOM fluxes using benthic incubation chambers is independent of such uncertainties. This approach is based on the assumption that solutes, released into the benthic chamber, behave conservatively throughout incubation, and, show linear trends over time.

It was previously suggested that porewater DOM consists in part of recalcitrant low molecular weight (LMW) compounds (Burdige and Gardner, 1998; Burdige and Komada, 2015). Therefore, the sediment outflux of DOM was hypothesised to serve as an important source of recalcitrant DOM to the water column (e.g. Burdige and Komada, 2015; Burdige et al., 2016). At the same time, elevated concentrations of dissolved organic nitrogen (DON) suggest the presence of labile proteinaceous organic matter in the porewaters (e.g. Faganeli and Herndl, 1991). Furthermore, measurements and modelling of isotopic

carbon composition in the anoxic and suboxic sediments off California suggest that about 50 % of DOM within the upper

sediment porewaters are represented by isotopically young and labile DOM components, that may be released to the water column, where they are actively utilised by heterotrophs (Bauer et al., 1995; Komada et al., 2013; Burdige et al., 2016).

Similarly to DOM in the water column, porewater DOM consists of a complex mixture of organic components, only a little fraction of which may be characterised by chemical analyses (e.g. Burdige and Komada, 2015). Therefore, examining the elemental composition of DOM or its optical properties may be useful for accessing the quality and reactivity of porewater DOM. The elemental ratio (DOC/DON) that is commonly used for inferring organic matter bioavailability in the water column, displays controversial patterns in sediment porewaters. Some studies suggest that low DOC/DON ratios of 2 to 5 found in sediments with reduced $O_2$ levels, may indicate an accumulation of bioavailable DOM under low $O_2$ conditions (Faganeli and Herndl, 1991; Alkhatib et al., 2013). On the other hand, DOC/DON ratios found in other studies were lower under oxygenated conditions compared to those of anoxic sediments (Burdige and Gardner, 1998).

Optical properties were also shown to provide important insights into DOM cycling not only in the water column (e.g. Coble, 1996; Zsolnay et al., 1999; Jørgensen et al., 2011; Catalá et al., 2016; Loginova et al., 2016) but also in porewaters of marine and freshwater sediments (e.g. Chen et al., 2016). The fraction of DOM that exhibits optical activity owing to the presence of chromophoric groups — a combination of conjugated double bonds and heteroatoms — in its molecular structure is referred to as chromophoric DOM (CDOM) and fluorescent DOM (FDOM). CDOM refers to DOM that absorbs light over a broad spectrum region, from UV to visible wavelengths. A typical CDOM absorbance spectrum is shaped as an exponential curve (Del Vecchio and Blough, 2004). The spectral slope ($S$) and absorption coefficients are used to learn on bulk DOM properties. For instance, a decrease in module value of $S$ may indicate an increase in relative molecular weight (e.g. Helms et al., 2008). Those changes in optical properties occur due to the ability of high molecular weight (HMW) DOM to absorb light at longer wavelengths, compared to LMW-DOM. Some fraction of CDOM is fluorescent and is mainly associated with aromatic molecular structures. This part of DOM is referred to as FDOM and is used to infer DOM quality (Coble, 1996; Zsolnay et al., 1999; Jørgensen et al., 2011; Catalá et al., 2016; Loginova et al., 2016). Thus, 3D fluorescence spectroscopy, followed by parallel factor analysis (PARAFAC), has been recognised as a useful tool for distinguishing between different organic matter pools (Murphy et al., 2013). Fluorophores that are excited and emit at UV wavelengths are often referred to as amino acid-like DOM. Components that are excited at UV, but emit at visible wavelengths, are mainly referred to as humic-like or fulvic-like DOM (e.g. Coble, 1996; Murphy et al., 2014, and references therein). CDOM distributions in sediment cores from the Chukchi Sea suggested that anoxic sediments serve as a production site of humic-like substances and a potential source of altered DOM for the water column (Chen et al., 2016). In turn, FDOM measurements made during incubations of sediment cores (Yang et al., 2014), indicated that DOM released into the overlying water might be further altered by microbial communities, serving as a source of bioavailable organic matter. In the ETSP off Peru, fine-scale spatial resolution FDOM measurements suggested DOM release from anoxic sediments into the water column (Loginova et al., 2016). The high FDOM fluorescence associated with the benthic release of DOM even reached the euphotic zone, likely influencing organic carbon turnover of the whole water column. The sediment release of DOM could potentially serve as an important carbon and N source (e.g. Moran and Zepp, 1997), and reduce the penetration depth of light in the water column (e.g. Belzile et al., 2002), potentially affecting phototrophic pelagic microbial communities, hence influencing biogeochemical processes of the water

column. However, still little is known about the release of porewater DOM and its reactivity in particular in the ETSP off Peru. In this study, we combine measurements of diffusive and in situ net fluxes of DOC and DON, and interpret those fluxes in relation to DOM optical properties. Our objectives are to provide a deeper understanding of DOM cycling in Peruvian sediments and across the sediment-water interface.

## 2   METHODS

### 2.1   Study area

Sediment sampling was carried out in April-May 2017 during the research cruises M136 and M137 to the Peruvian OMZ on board of RV Meteor. The sampling area was located between 12-12.2 °S and 77.1-77.3 °W (Fig. 1). In total, six stations were sampled along the transect 12°S (12°S) (see Table 1) on the middle shelf, outer shelf, and continental slope (Dale et al., 2015, 2016; Sommer et al., 2016).

During the study, the water column at the sampling stations was subjected to a consistent poleward flow ranging from 0.1 to 0.5 $\mathrm{m\,s^{-1}}$ (Lüdke et al., 2019). Low-oxygen ($\ll 5\,\mathrm{\mu mol\,kg^{-1}}$) waters were observed above the sediment, with the exception for station 2 (St.2), where the $O_2$ concentration was slightly above 10 $\mathrm{\mu mol\,kg^{-1}}$. This may have been a remnant of the coastal el Niño that occurred 3–4 months prior to our fieldwork (Rodríguez-Morata et al., 2019) or intensification of poleward flow, observed in May 2017 (Lüdke et al., 2019). High concentrations of water column nitrate ($NO_3^-$) and nitrite ($NO_2^-$) were observed at stations deeper than 100 m water depth, while at shallower stations ammonium ($NH_4^+$) was dominant dissolved inorganic N component. Thus, $NH_4^+$ concentrations up to 1.2–1.4 $\mathrm{\mu mol\,L^{-1}}$ were detected in the middle shelf stations (Lüdke et al., 2019).

Sediments at the sampling stations are fine-grained diatomaceous dark-olive anoxic muds (Gutiérrez et al., 2009; Mosch et al., 2012) with porosities ranging between 0.8 and 0.9 (Table 1). In previous studies, polychaetes and oligochaetes were found in the sampling area (Dale et al., 2015; Sommer et al., 2016). However, the sediment showed little evidence of strong mixing by bioturbation (Bohlen et al., 2011; Dale et al., 2015). Instead, the sediments were densely colonised by mats of large filamentous sulphur bacteria of the genera *Thioploca spp.* and *Beggiatoa spp.* (Gutiérrez et al., 2009; Mosch et al., 2012). Dale et al. (2015) reported that mats of these sulphide oxidising bacteria cover up to 100 % of the sediment surface at the shallower stations extending their trichomes 2 cm into the water column to access bottom water $NO_3^-$. They could be observed from the sediment surface down to 20 cm sediment depth. At the offshore stations, bacterial mats of several dm in diameter were covering up to 40 % of the sediment surface. Their occurrence was related to high organic carbon rain rates, which ranged from 10 $\mathrm{mmol\,m^{-2}d^{-1}}$ on the continental slope to 80 $\mathrm{mmol\,m^{-2}d^{-1}}$ on the shallowest shelf station (Fig.S1). Furthermore, the region was characterised by substantial organic matter utilisation as indicated by high DIC fluxes and porewater $NH_4^+$ concentrations (Dale et al., 2015). Thus, despite the high sediment accumulation rates and POC content of the sediments, the high organic matter respiration, as follows from large sediment DIC (Dale et al., 2015) and $NH_4^+$ (Sommer et al., 2016) fluxes at middle shelf stations, led to a low percentage of carbon burial efficiency (∼17%), compared to the outer shelf and the continental slope (24-74 %) (Fig.S1). Furthermore, Sommer et al. (2016) and Dale et al. (2015) suggested spatial

variability of biological N cycling in the area. Thus, outer shelf stations displayed the highest sediment uptake rate of $NO_3^-$ and $NO_2^-$ followed by high $N_2$ outflux (Sommer et al., 2016). At shallower stations, $NO_3^-$ and $NO_2^-$ were entirely exhausted and excessively high fluxes of $NH_4^+$ were observed (Sommer et al., 2016, Fig.S1). Those spatial variabilities in N fluxes were suggested to be a result of denitrification and anammox on the outer shelf and continental slope, and DNRA in the middle

shelf. A detailed description of the sediment and bottom waters at 12°S can be found in Dale et al. (2015, 2016) and Sommer et al. (2016).

## 2.2    Field sampling and sample preparation

Two benthic landers (Biogeochemical Observatory (BIGO) I and II) (Sommer et al., 2008) were deployed (see Table 1). Both were equipped with two circular flux chambers with an internal diameter of 28.8 cm. Volumes of the bottom water enclosed in

the benthic chambers varied from ∼12 to ∼20 L during the study. Each BIGO chamber was equipped with a stirrer and eight glass syringes, which were filled sequentially to determine the net *in situ* flux of solutes across the sediment-water interface (Fig. S2). A detailed description of the BIGO lander can be found in Sommer et al. (2008) and Dale et al. (2014).

At each station, water from one BIGO chamber (chamber 2) was used for the DOM sampling. Samples for DOC, DON, and CDOM and FDOM were taken at ∼0.2, 4, 9, 12, 17, 21, 25, and 30 hrs after the beginning of the sediment enclosure.

All samples were passed through pre-washed (60 mL of ultrapure water) cellulose acetate (CA) membrane syringe filters (0.2 μm). The first 5 mL of the filtrate was discarded to waste before filling the sample into storage vials. Several types of filters (PES, nylon, CA, and regenerated cellulose (RC)) were tested for potential DOC and total dissolved nitrogen (TDN) contamination before the cruise. CA and RC filters produced minimal background concentrations for both parameters after rinsing with 60 ml of ultrapure water (see Fig.S3). CA filters were chosen over RC due to their lower binding affinity to

macromolecules and proteins.

Filtered samples were filled into pre-combusted (450°C, 8 hrs) amber glass vials for CDOM and FDOM and into pre-combusted (450°C, 8 hrs) clear glass ampoules for DOC and DON analyses. The latter samples were fixed with 20 μl of ultra-pure HCl (30 %: Merck Chemicals GmbH) and flame sealed before storage. All samples were stored (1-2 month) at +4 °C in the dark pending analysis in the home laboratory.

The porewater DOM distribution and properties, as well as diffusive fluxes, were quantified by analysing DOC, DON, CDOM and FDOM in sediment cores obtained using multicorers (MUCs). Retrieved sediments were immediately transferred to the cool onboard room (10-15 C°) and processed under anoxic conditions within a few hours using an argon-filled glove bag. One sediment core from each station was sectioned into 12 slices over intervals ranging from 1 to 3 cm (Fig. S2). Sediments were transferred into acid-cleaned (10 % HCl) dry polypropylene (50 ml) centrifugation tubes and spun in a refrigerated

centrifuge for 20 min at 4500 rpm. The supernatant was then passed through cellulose acetate membrane syringe filters (0.2 μm) into pre-combusted (450°C 8 hrs) clear glass ampoules for DOC and DON and amber glass vials for CDOM and FDOM. The samples were conserved and stored as described above.

Studies conducted in areas with abundant macrofauna suggested that pore waters isolated by centrifugation exhibit higher DOC concentrations compared to non-invasive methods, such as so called sip-isolation (Alperin et al., 1999). Macrofauna cell

rupture during centrifugation was suggested to influence the extracted DOC, and the removal of macrofauna from sediments before centrifugation and whole-core squeezing was shown to reduce elevated DOC concentrations (Martin and McCorkle, 1993). Our study site did not exhibit signatures of significant bioturbation (Dale et al., 2015). Accordingly, DOC concentrations at sites similar to our study area (low oxygen - low bioturbation), which were extracted by centrifugation showed

agreement either with those obtained by sip-isolation method (Komada et al., 2004) or with those obtained from *in situ* and *ex situ* incubations (Holcombe et al., 2001). Furthermore, Holcombe et al. (2001) suggested that sip-isolated porewater DOC gradients may lead to underestimation of diffusive DOC fluxes in low-bioturbation regions. Thus, varying strength of organic matter–mineral associations may create different solute reservoirs around the surface of a mineral. The sip-isolation method was suggested to extract only loosely bound DOM out of the marine sediments, while centrifugation would sufficiently perturb

sediments and sample the majority of the porewater DOM that may efflux out of the sediment. In connection with the above, the centrifugation method was preferred as a pore water extraction method for DOM analyses in this study.

## 2.3 Discrete sample analyses

CDOM absorbance was measured with a Shimadzu® 1700 UV-VIS double-beam spectrophotometer using a 1-cm Quartz SUPRASIL® precision cell (Hellma® Analytics). Absorbance spectra were recorded at 1 nm wavelength intervals from 230

to 750 nm against MilliQ water as a reference. CDOM absorbance spectra in the wavelength interval from 275 to 400 nm were corrected for particle scattering (e.g. Nelson and Siegel, 2013) and recalculated to absorption, according to Bricaud et al. (1981). This method has a detection limit of $\sim$0.001 absorption units (that may be referred to $\sim$0.5 m$^{-1}$) and a precision <5%, estimated as the maximal standard deviation of CDOM absorbance spectra in the wavelength interval from 275 to 400 nm divided by the mean value of three repeated measurements. We used the absorption coefficient at 325 nm ($a_{CDOM}(325)$) to

express CDOM "concentrations", as this one is mainly used for open ocean areas (Nelson and Siegel, 2013). The spectral slope ($S$) for the interval 275-295 nm ($S_{275-295}$) was used to infer relative changes in DOM bulk quality, i.e. DOM relative molecular weight (Helms et al., 2008). $S_{275-295}$s were calculated following Helms et al. (2008) using log-transformed linear regression.

FDOM was analysed by Excitation-Emission Matrix (EEM) spectroscopy on a Cary Eclipse Fluorescence Spectrophotometer (Agilent Technologies) equipped with a xenon flash lamp. The fluorescence measurements for samples were done in a 4-

optical window 1-cm Quartz SUPRASIL® precision cell (Hellma®Analytics). Blank and Water Raman scans were performed daily using an Ultra-Pure Water Standard sealed cell (3/Q/10/WATER; Starna Scientific Ltd). The experimental wavelength range for sample scans and ultra-pure water scans was 230 to 455 nm in 5 nm intervals on excitation and 290 to 700 nm in 2 nm intervals on emission. Water Raman scans were recorded from 285 to 450 nm at 1 nm intervals for emission at the 275 nm excitation wavelength (Murphy et al., 2013). All fluorescence measurements were conducted at 20 °C, controlled by a Cary

Single Cell Peltier Accessory (VARIAN), PMT 900 V, with 0.2 s integration times and a 5 nm slit width on excitation and emission monochromators. The precision of this method does not exceed 3% if estimated as a standard deviation of Raman peaks at 275 nm of each measurement day, divided by the mean value. The fluorescence EEMs were corrected for spectral bias, background signals and inner filter effects and normalised to the area of ultra-pure water Raman peaks. All samples were calibrated against a Quinine Sulphate Monohydrate dilution series, performed once during sample analyses. EEMs were anal-

ysed by PARAFAC (Stedmon and Bro, 2008) and validated by split-half analysis using "drEEM toolbox for MATLAB" after Murphy et al. (2013). Four FDOM components that were identified during the PARAFAC analyses are expressed in Quinine Sulphate Equivalents (QSE).

Samples for inorganic N compounds in the benthic lander samples ($NO_3^-$, $NO_2^-$ and $NH_4^+$) and the porewaters ($NH_4^+$) were analysed following standard techniques after Hansen and Koroleff (2007) and will be published elsewhere (Clemens et al., in prep.). $NO_3^-$ and $NO_2^-$ concentrations in the porewaters were assumed to be negligible (Dale et al., 2016) and not analysed. Detection limits for the determination of $NO_3^-$, $NO_2^-$ and $NH_4^+$ were 0.05, 0.01, and 0.5 $\mu mol\,L^{-1}$, respectively.

DOC samples were analysed by the high-temperature catalytic oxidation (TOC -VCSH, Shimadzu) with a detection limit of 1 $\mu mol\,L^{-1}$, as described in detail by Engel and Galgani (2016). Calibration of the instrument was performed every second week using six standard solutions of 0, 500, 1000, 1500, 2500 and 5000 $\mu gC\,L^{-1}$, which were prepared using a potassium hydrogen phthalate standard (Merck 109017). Before each set of measurements, a baseline of the instrument was set using ultrapure water. The deep-sea standard (Dennis Hansell, RSMAS, University of Miami) with known DOC concentration was measured after setting the baseline to verify accuracy by the instrument. Typically, the precision of the method did not exceed 4 %. Furthermore, two control samples with known concentrations of DOC were prepared for each day of measurement using a potassium hydrogen phthalate standard (Merck 109017). The DOC concentrations of those control samples were typically within the range of samples' concentrations and were measured at the time of sample analyses to control baseline flow during measurements. The DOC concentration was determined in each sample out of five to eight replicate injections.

A TNM-1 N detector of the Shimadzu analyser was used to determine TDN in parallel to DOC with a detection limit of 2 $\mu mol\,L^{-1}$ (Dickson et al., 2007). Calibration was performed simultaneously with the calibration of carbon detector using standard solutions of 0, 100, 250, 500 and 800 $\mu g\,N\,L^{-1}$, which was prepared using potassium nitrate Suprapur (Merck 105065). The deep-sea standard (Dennis Hansell, RSMAS, University of Miami) with the known concentration of TDN was measured daily to verify the accuracy of the instrument. The precision of the method did not exceed 2 % estimated as the standard deviation of 5–8 injections divided by the mean value. Concentrations of DON were calculated as a difference of TDN and the sum of concentrations of inorganic N components. The differences of analytical methods for the determination of TDN and dissolved inorganic N species, particularly in systems dominated by dissolved inorganic N, may induce negative values during the quantification of DON (Westerhoff and Mash, 2002). In this case, DON concentrations were set to "0" and, therefore, were excluded from calculations of DOC/DON ratios. In the text, those values were presented as "below detection limit (b.d.l.)".

## 2.4 Evaluation of DOC and DON fluxes

In this study, diffusive and *in situ* net DOC and DON fluxes were quantified. The initial concentrations in BIGO chambers and porewater solute concentrations from the uppermost slice of the sediment core (0 to 1 cm depth) were used for the flux calculations. Thus, the diffusive fluxes of DOC ($J_{DOC}(Diff.)$) and DON ($J_{DON}(Diff.)$) were estimated by applying Fick's First Law:

$$J_s(Diff.) = -\phi \times D_s \times \frac{dC}{dz} \tag{1}$$

where $J_s(Diff.)$ is a diffusive flux of a solute, $\phi$ is the sediment porosity, $\dfrac{dC}{dz}$ is the gradient of DOC (DON) concentration over the investigated depth interval (0 to 1 cm), and $D_s$ is a bulk sediment diffusion coefficient. $D_s$ was previously demonstrated to be dependent on the sediment formation resistivity factor ($F$) (Ullman and Aller, 1982), as well on the average molecular weight of DOM (Burdige et al., 1992; Balch and Guéguen, 2015). In this study, we calculate $D_s$ using $F$ that equals $\phi^{-3}$

(Ullman and Aller, 1982), as $\phi$ measured at 12°S exceeded 0.8-0.9 (Table 1). The molecular size fractionation was not addressed during this study, therefore, we assumed that DOM molecular weight varied in the range from 0.5 to 10 kDa. This assumption resulted in $D_0$ varying from $0.63 \times 10^{-6}$ to $7.2 \times 10^{-6}$ $\mathrm{cm^{-2}\,s^{-1}}$ (Balch and Guéguen, 2015). This variance represented one of the major drivers of the estimated diffusive DOC (DON) flux variability. Therefore, the calculation of $J_s(Diff.)$ was done for the the whole range of $D_0$ with an increment of $0.1 \times 10^{-6}$. Thus, $J_s(Diff.)$ presented in this manuscript is a resulting

mean value of all the calculated $J_s(Diff.)$, and its variability expressed as a standard deviation.

Net *in situ* fluxes of DOC ($J_{DOC}(Net)$) and DON ($J_{DON}(Net)$), measured in BIGO chambers, were evaluated as:

$$J_s(Net) = \frac{V}{A} \times \frac{dC}{dt} \tag{2}$$

where $J_s(Net)$ net *in situ* flux of a solute, $V$ is the chamber volume (in $\mathrm{m^3}$), $A$ is the chamber area (in $\mathrm{m^2}$), and $\dfrac{dC}{dt}$ is the DOC (DON) concentration gradient over the time of the sediment enclosure (in $\mathrm{mmol\,m^{-3}d^{-1}}$). The gradient was obtained

by linear regression analyses ('polyfit' 1st order, MatLab, The MathWorks Inc.) of the DOC (DON) concentrations over time. The error of the linear regression was used as a representation of the standard deviation of the evaluated net fluxes.

In this study, fluxes directed out and into the sediment are reported as positive and negative values, respectively.

## 3  RESULTS

### 3.1  DOC and DON distribution and fluxes

Porewater DOC generally accumulated with depth in the sediment (Fig.2). The highest concentrations of DOC were measured at the middle shelf at station 1 (St.1), ranging from 152 $\mathrm{\mu mol\,L^{-1}}$ at 0.5 cm to a maximum of 2.6 $\mathrm{mmol\,L^{-1}}$ at 22.5 cm of sediment depth. Porewater DOC concentrations and gradients decreased gradually towards station 4 (St.4), where DOC concentrations ranged from 122 $\mathrm{\mu mol\,L^{-1}}$ at 0.5 cm to 544 $\mathrm{\mu mol\,L^{-1}}$ at 22.5 cm of sediment depth. Further offshore, porewater DOC concentrations and gradients increased at station 5 (St.5) and station 6 (St.6), ranging from 177 $\mathrm{\mu mol\,L^{-1}}$ at 0.5 cm to 823

$\mathrm{\mu mol\,L^{-1}}$ at 22.5 cm and from 210 $\mathrm{\mu mol\,L^{-1}}$ at 1.5 cm to 702 $\mathrm{\mu mol\,L^{-1}}$ at 19.5 cm, respectively. The highest concentrations of DON were measured at the middle shelf St.1 and St.2 (Fig.2, Fig.S4. Fig. S5). The DON concentrations in porewaters at these stations were ranging from b.d.l. at 0.5 cm to a maximum of 2.6 $\mathrm{mmol\,L^{-1}}$ at 22.5 cm and from 580 $\mathrm{\mu mol\,L^{-1}}$ at 0.5 cm to 1.1 $\mathrm{mmol\,L^{-1}}$ at 19.5 cm of sediment depth, respectively. Similarly to DOC, porewater DON concentrations decreased towards St.4, where they ranged from b.d.l. at surface sediment to 85 $\mathrm{\mu mol\,L^{-1}}$ at 3.5 cm sediment depth and then increased

again at St.5 (64–450 $\mathrm{\mu mol\,L^{-1}}$) and St.6 (b.d.l.–248 $\mathrm{\mu mol\,L^{-1}}$).

In general, sediment porewaters at 12ºS exhibited low DOC/DON ratios. Generally, the median elemental ratio increased towards offshore from the minimum at St.2 (DOC/DON of <1) to the maximum at St.4 (median DOC/DON ∼12) and then decreased again at St.5 (median DOC/DON ∼1) and St.6 (median DOC/DON ∼3) (Fig.S6).

No apparent differences in DOC concentrations within benthic chambers were observed between stations (Fig. 3). The average concentrations for all the incubation chambers over time were $92\pm22\,\mu mol\,L^{-1}$. Furthermore, DOC did not accumulate linearly over time at some stations (Fig.3). Similarly, DON concentrations varied from b.d.l. to ∼15 $\mu mol\,L^{-1}$ in the chambers (Fig.3), resulting in much higher DOC/DON ratios than measured in the porewaters. Median DOC/DON ratios in all chambers calculated over time were >5, gradually decreasing from a maximum at St.1 (median DOC/DON ∼30) towards median DOC/DON ∼8.5 offshore (Fig.S6).

The diffusive DOC fluxes on the outer shelf and continental slope stations varied from a minimum of $0.2\pm0.1\,mmol\,m^{-2}d^{-1}$ at St.4 to a maximum of $2.5\pm1.3\,mmol\,m^{-2}d^{-1}$ at station 3 (St.3) (Fig. 4). Net *in situ* DOC fluxes determined with benthic chambers were lower than diffusive fluxes on those stations, varying from -0.3±0.9 at St.4 to 0.8±0.9 $mmol\,m^{-2}d^{-1}$ at St.3. Net *in situ* DOC fluxes on the middle shelf stations were higher than fluxes estimated by Fick's law. Thus, the diffusive DOC fluxes were varying from 0.2±0.1 to 0.4±0.2 $mmol\,m^{-2}d^{-1}$ and net *in situ* DOC were ranging between 1.1±0.9 and 2.3±2.3 $mmol\,m^{-2}d^{-1}$. Diffusive DON fluxes ranged from -0.04±0.02 $mmol\,m^{-2}d^{-1}$ at St.1 and St.6 to 3.3±1.7 $mmol\,m^{-2}d^{-1}$ at St.2. Similar to DOC, net *in situ* DON fluxes were lower than diffusive DON fluxes on the outer shelf and continental slope stations, ranging from -0.05±0.3 $mmol\,m^{-2}d^{-1}$ at St.6 to 0.3±0.3 $mmol\,m^{-2}d^{-1}$ at St.5. In contrast to DOC fluxes, the diffusive DON flux on one of the middle shelf stations (St.2) was also higher than the net *in situ* DON flux, exhibiting 3.3±1.7 $mmol\,m^{-2}d^{-1}$ and -0.03±0.3 $mmol\,m^{-2}d^{-1}$, respectively. At St.1 both diffusive and net *in situ* DON flux estimates were very low. They displayed -0.04 ±0.02 $mmol\,m^{-2}d^{-1}$ and 0.08±1.4 $mmol\,m^{-2}d^{-1}$, respectively. Despite the clear apparent distinction between the different flux estimates for both, DOC and DON, no statistical differences were found between them at each station ($p$>0.05, Mann-Whitney Rank Sum Test, SigmaPlot, Systat Software).

## 3.2   Optical properties of DOM

To address DOM quality CDOM and FDOM fluorescence intensities were analysed from the sediment porewaters and the BIGO chambers.

In the porewaters, CDOM absorption coefficients ($a_{CDOM}(325)$) exhibited a similar pattern to DOC distribution (Fig.2). The highest values of $a_{CDOM}(325)$ were measured at St.1. They ranged from 3.2 $m^{-1}$ at 0.5 cm to 22.8 $m^{-1}$ at 22.5 cm of sediment depth. The lowest values of $a_{CDOM}(325)$ were measured at St.4, ranging from 2.7 $m^{-1}$ at 0.5 cm to 8.9 $m^{-1}$ at 7 cm of sediment depth. Further offshore, at St.5 and St.6 values of $a_{CDOM}(325)$ were higher than at St.4.

In the benthic chambers, at the outer shelf and continental slope, $a_{CDOM}(325)$s generally ranged from 0.3 to 2.5 $m^{-1}$ (Fig.3), exhibiting, however, different trends. Thus, an apparent decrease in $a_{CDOM}(325)$ over time occurred at St.3, St.5 and St.6, while at St.4 $a_{CDOM}(325)$ dynamics suggested an apparent accumulation of CDOM. The middle shelf stations, St.1 and St.2, displayed lower variance, ranging from 0.1 to 1 $m^{-1}$ over time, and exhibited no visible trends (Fig.3, Table S1).

The CDOM spectral slope, $S_{275-295}$, in the porewaters, increased with depth in all sediment cores, displaying the highest values (-0.016±0.004 $nm^{-1}$) at St.4, and the lowest values at St.1 $S_{275-295}$ (-0.018±0.001 $nm^{-1}$). The latter values were comparable to the initial values of $S_{275-295}$ in the BIGO benthic chambers (-0.018±0.005 $nm^{-1}$) (see Fig.2 and Fig.3).

In the BIGO chambers, the highest $S_{275-295}$ were observed at the beginning of the sediment enclosure, and an apparent $S_{275-295}$ decrease occurred over time (Fig. 3). The decrease in $S_{275-295}$ was steeper at stations with higher porewater DOC content. Thus, the fastest change in $S_{275-295}$ occurred at St.1 (-0.016±0.017 $nm^{-1}d^{-1}$) whereas the slowest change was found at St.4 (-0.004±0.006 $nm^{-1}d^{-1}$). (Fig.3, Table S1).

FDOM spectroscopy and PARAFAC analyses allowed four independent fluorescent components to be distinguished (Fig.5). FDOM components that are excited at UV and emit in the visible wavelength range were previously referred to as humic-like substances (e.g., Coble, 1996; Murphy et al., 2013, 2014; Loginova et al., 2016, and references therein). Here, two fluorescent components, FDOM component 1 (Comp.1) and FDOM component 2 (Comp.2), with excitation and emission (Ex/Em) of 370/464 nm and 290-325/400 nm, respectively, were identified and referred to as humic-like components (Fig. 5). Amino acid-like substances are the second group of well-determined FDOM components (e.g., Coble, 1996; Murphy et al., 2013, 2014; Loginova et al., 2016, and references therein) corresponding to molecules that are excited and emit in the UV wavelength range. Thus, FDOM component 3 (Comp.3) and FDOM component 4 (Comp.4), with Ex/Em of 290/340(684) nm and 275/310(600) nm, respectively, were assumed to represent proteinaceous DOM (Fig. 5). During this study, humic-like components showed similar trends to DOC and $a_{CDOM}(325)$ in the porewaters. Their fluorescence intensified with sediment core depth and decreased offshore with a minimum fluorescence at St.4 (Fig.6). Amino acid-like Comp.3 and Comp.4, also increasing in the porewaters with depth, but were generally depleted throughout the sediment except for St.1, where their fluorescence reached the max. 6 QSE and max. 1.7 QSE, respectively (Fig.6).

In the benthic chambers, all fluorescent component QSEs were nearly an order of magnitude lower than those in the porewaters. An apparent increase within the chambers was observed in the humic-like Comp.1 and Comp.2 and the amino acid-like Comp.4 (Fig.7).

At St.1, St.3, St.4 and St.6, the Comp.3 was slightly enhanced at the beginning of the sediment incubation followed by an apparent removal at a later stage. Among nearly all stations, the humic-like Comp.1, Comp.2 and amino acid-like Comp.4 displayed similar gradients of ∼0.03, 0.06–0.08 and 0.03–0.04 $QSE\,d^{-1}$, respectively. Exceptions were observed at St.4, which displayed Comp.1,Comp.2 and Comp.4 gradients of 0.001, 0.04 and -0.005 $QSE\,d^{-1}$, respectively; and St.1, where the gradients of Comp.2 and Comp.4 were ∼0.04 and ∼0.09 $QSE\,d^{-1}$, respectively (Table S1).

## 4 DISCUSSION

### 4.1 Spatial variability of the DOM fluxes along the 12ºS transect

Spatial variability of organic matter decomposition in sediments is a common feature in the world ocean (see Arndt et al., 2013, for an overview). This variability is naturally attributed to the efficiency of vertical transfer of POM to the sediment (e.g. Seiter et al., 2004; Marsay et al., 2015; Engel et al., 2017). Along 12ºS, highest sedimentation rates, estimated via $^{210}Pb_{xs}$

activity, were reported for the middle shelf St.1 and St.2 (Fig. S1), while St.4 displayed the lowest sedimentation rates and porewater DOM concentrations possibly caused by an inhibition of particle settling by bottom currents (Dale et al., 2015). The highest accumulation of POM along 12°S was also observed at St.1 and St.2 (Fig. S1) even though the organic carbon burial efficiency exhibited lower values at the middle shelf stations than at the stations offshore (Dale et al., 2015). Lower carbon

burial efficiency in combination with very high rates of organic matter remineralisation, as follows from extremely high DIC and $NH_4^+$ benthic fluxes (Dale et al., 2015; Sommer et al., 2016) (Fig. S1), suggests higher bioavailability of POM supplied to the middle shelf. Accordingly, porewater DOM optical properties reflected the "freshest" character of organic matter at St.1 and St.2, where $S_{275-295}$ values were similar to those in the water column (Fig. 3) and protein-like DOM fluorescence (Fig. 6) and DON were highly enriched (Fig. 3). Therefore, and in line with the previous findings, our data suggest that the middle shelf

stations are supplied with more labile POM compared to the outer shelf stations. This labile POM, likely of proteinaceous origin (e.g. Faganeli and Herndl, 1991), is, in turn, rapidly reworked. Thus, in the middle shelf St.1, despite the highest accumulation of POC (Dale et al., 2015) and elevated porewater DOC and especially DON concentrations (Fig.2, Fig.S4), the diffusive fluxes of DOC and DON here were not highest on the transect (Fig.4). Since high values of $a_{CDOM}(325)$ and protein-like FDOM have previously been related to labile DOM (Loginova et al., 2016), one may expect DOM with such characteristics to being rapidly

reworked by heterotrophic microbial communities. Therefore, proteinaceous FDOM and $a_{CDOM}(325)$ nonlinear distribution over time might be a result, as of the low fluxes from the sediment, but also the signature of its rapid microbial utilisation (Komada et al., 2016). The strong decrease in $S_{275-295}$ and the accumulation of humic-like substances observed at St.1 during the incubations point to both, a high benthic release of fresh bioavailable DOM at St.1 as well as its rapid consumption and reworking at the sediment-water interface. These results support the idea that microbial utilisation is controlled by the quality

of the supplied organic matter (Pantoja et al., 2009; Le Moigne et al., 2017). At the same time, spatial variability of benthic fluxes could also be attributed to the spatially variable DOM recycling efficiencies of different biogeochemical processes. For instance, denitrification and anammox were found to be the major processes of N cycling in the outer shelf and on the upper continental slope, whereas the middle shelf stations, had elevated rates of DNRA (Dale et al., 2016; Sommer et al., 2016). While the linkages between microbial N turnover and DOM fluxes still not well understood, it is noteworthy that the middle shelf

sediments were covered with *Marithioploca* mats that greatly affect the N and sulphur biogeochemical cycles and, potentially, DOM cycling and reactivity.

At St.2, DON accumulated to higher levels within the porewaters than DOC and $NH_4^+$ (see e.g. Fig. S4), leading to higher diffusive DON fluxes than those of DOC and to extremely low DOC/DON ratios (Fig. S6). In agreement to this $S_{275-295}$ revealed lowest changes over time, suggesting that at St.2 DOM in the benthic chamber remained "fresher" during the incubation, compared to other stations. Similar to that, proteinaceous Comp.3, despite its generally low variability, exhibited the highest

increase at St.2, suggesting a relative accumulation of proteinaceous DOM in the corresponding chamber. Herewith, porewater DON concentrations generally seemed to be more responsive to fresh organic matter input (Dale et al., 2015), while DOC, accumulated more steadily during organic matter degradation, as indicated by $NH_4^+$ concentrations (Fig.S4). Those results are in line the "decoupling" between DOC and DON remineralisation as previously suggested by e.g. Alkhatib et al. (2013). These

authors suggested that the enzymatic hydrolysis of N-containing labile POM occurs at a higher rate than that of carbon-rich

compounds, leading to a higher accumulation of DON over DOC in the porewaters. Furthermore, the dissolved by-products of bacterial activity are often found to be enriched in N, and therefore sediments with pronounced microbial activity show relatively low DOC/DON ratios (Burdige and Komada, 2015). For instance, glycine (DOC/DON=2) was suggested to preferentially accumulate as a result of microbial metabolism in mixed redox sediments (Burdige, 2002). Bioturbation by macro-biota

in oxygenated sediments is also often associated with the accumulation of urea (DOC/DON=0.5) (Burdige and Gardner, 1998). However, given that retrieved sediment cores were not bioturbated, active remineralisation of bioavailable organic matter by microbial communities within the sediment is more likely. Besides, DOM itself may enter chemical reactions with hydrogen sulphide that is produced in large quantities at middle shelf stations (Schunck et al., 2013; Sommer et al., 2016). For instance, quinone structures can react with hydrogen sulphide, producing hydroquinones (Heitmann and Blodau, 2006), which may be

further utilised by methanogenic degradation processes (Szewzyk et al., 1985). This could affect DOC and DON porewater concentrations and a decrease in the diffusive DOC flux over the diffusive DON flux. However, the extreme accumulation of DON over the DOC in porewaters at St.2 and also St.1 seems to be hardly explainable with the organic N sources alone. Herewith, the actual mechanisms behind the decoupling of DOC and DON fluxes remain obscure.

## 4.2   Porewater DOM utilisation at the sediment-water interface

In current understanding, production of DOM from POM degradation processes followed by microbial utilisation of DOM (e.g. Burdige and Komada, 2015) can cause an imbalance in DOM production and consumption, resulting in a net accumulation of DOM with sediment depth. This is in part explained by an accumulation of recalcitrant DOM, which is thought to be of LMW, in the sediments produced as a result of the "microbial carbon pump" (Burdige and Komada, 2015). Furthermore, physico-chemical processes, such as: 1) irreversible sorption onto particles, 2) aggregation (Liu and Lee, 2007; Arndt et al., 2013),

3) reactions of chelation and 4) co-precipitation (Lalonde et al., 2012), or 5) inhibition of microbial activity (Emerson, 2013; Canfield, 1994; Aller and Aller, 1998) may affect the DOM accumulation in sediment porewaters. At the same time, measurements of $\Delta^{14}C$ in the porewater DOM suggests that its substantial fraction is isotopically young and is readily utilised by heterotrophic communities, when released to the water column (Bauer et al., 1995; Komada et al., 2013; Burdige et al., 2016). The observed accumulation of porewater DOM with depth (Fig. 2) agrees well with previous observations (Burdige and Gard-

ner, 1998; Komada et al., 2004; Chipman et al., 2010; Alkhatib et al., 2013) as well as with DOC concentrations reported for non-bioturbated anoxic sediments ($\sim$1-3 mmol l$^{-1}$) (Burdige and Komada, 2015). The increase of humic-like fluorescence and its correlation with DOC concentrations (Comp.1, R=0.8, n=0.86, $p$<0.01), as observed during our study, has also been noted previously in marine sediments (e.g. Chen et al., 1993) and is commonly explained as a net production of LMW recalcitrant humic DOM (Komada et al., 2004). The increase in S$_{275\text{-}295}$ over sediment depth indicated an increase in apparent molecular

weight (Helms et al., 2008). This apparent increase in molecular weight in combination with the down-core intensification in humic-like fluorescence may, therefore, suggest polymeric LMW (pLMW) DOM formation. This may undergo via reactions of polymerisation (Hedges et al., 1988) or complexation (Finke et al., 2007), as well as due to the formation of supramolecular clusters via hydrogen bonding or hydrophobic interactions (e.g. Sutton and Sposito, 2005). The down-core accumulation of DON and of amino acid-like FDOM, and also the correlation of amino acid-like FDOM to DOC (Comp.4, R=0.6, n=0.86,

$p$<0.01) suggest that proteinaceous DOM is also being produced during POM remineralisation in sediments. Given that the second emission peaks of Comp.3 and Comp.4 displayed spectral characteristics similar to chl *a* and its auxiliary carotenoids (e.g. Wolf and Stevens, 1967), the protein-like FDOM components are likely products of phytoplankton debris recycling.

Benthic DOM fluxes were previously shown to contribute an important fraction to the organic matter that escapes remineral-
isation in the sediments (e.g. Ludwig et al., 1996; Burdige et al., 1999). Net *in situ* benthic DOC fluxes found during our study (-0.3±0.9–2.3±2.3 $mmol\,m^{-2}d^{-1}$) (Fig. 4) were comparable to previous estimates for shelf and continental slope sediments off the coasts of Peru and California, ranging from 0.03–4.41 $mmol\,m^{-2}d^{-1}$ (see Burdige et al., 1992, 1999; Burdige and Komada, 2015, for full overview). However, a linear accumulation of DOC and DON in benthic chambers (Burdige et al., 1992; Burdige and Homstead, 1994; Burdige et al., 1999) over time was generally not observed. We were able to trace the qualitative
transformations of DOM in benthic chambers over the investigated period by the changes in DOM optical properties. The decrease in $S_{275\text{-}295}$ and an intensification of humic-like fluorescence over time indicated an accumulation of LMW humic DOM components (Helms et al., 2008). At the same time, the complex development of the amino acid-like fluorescence of Comp.3 and the drawdown of $a_{CDOM}(325)$ and DON, resulting in increased DOC/DON ratios, suggested a reworking of proteinaceous labile DOM. Therefore, the production of humic-like LMWDOM along with the utilisation of proteinaceous DOM suggests an
active microbial DOM utilisation occurring at the sediment-water interface. These results support the idea that DOM release to the water column may stimulate respiration by water column microbial communities (Alkhatib et al., 2013; Komada et al., 2013; Burdige et al., 2016).

As stated previously, the rate of organic matter decomposition in sediments may depend not only on organic matter bioavailability (Canfield, 1994), but also on the inhibition of microbial activity (Aller and Aller, 1998), and the availability of electron
acceptors (Emerson, 2013; Canfield, 1994).

We suggest that the availability of electron acceptors, such as $NO_3^-$ and $NO_2^-$, in the water column above the sediments (Thomsen et al., 2016; Lüdke et al., 2019, and also Fig. S7) could stimulate microbial communities at the sediment-water interface to take up DOM.

Furthermore, the suggested formation of pLMWDOM in the sediment porewaters, due to geo-polymerisation, the forma-
tion of supra-molecules due to hydrogen bonding (Sutton and Sposito, 2005; Finke et al., 2007), or encapsulation by humic substances (e.g. Tomaszewski et al., 2011), might have reduced accessibility of bioavailable DOM compounds.

Labile substances, such as amino acids and carbohydrates, may have become unavailable for heterotrophic communities within the porewaters, resulting in DON accumulation with sediment depth. Herewith, the subsequent release of DOM into the water column may have lead to unfolding (solubilisation) of those supra-molecules due to, e.g. the chaotropic effect of $NO_3^-$
(e.g. Gibb and Gibb, 2011), and, consequently, an increase the DOM bioavailability for the microbial communities.

Therefore, a non-conservative behaviour of DOC and DON and proteinaceous FDOM in the BIGO chambers during sediment enclosure might be a result of sediment release/microbial DOM consumption and reworking at the sediment-water interface. Furthermore, DOM released by the sediment could potentially support an enhanced microbial abundance and carbon oxidation rates near the sediment along 12ºS transect (Maßmig et al., 2020) and influence the activity of microbial mats that
cover up to 100 % of the sediment surface at the middle shelf stations (Sommer et al., 2016). In turn, POM respiration rates,

which are commonly evaluated from DIC flux measured in benthic lander systems (Dale et al., 2015), may have been under-estimated, as the diffusive DOC fluxes, calculated in this study could represent up to $\sim 53\%$ of the estimated DIC flux ($J_{DIC}$, Clements et al., in prep.), and the net *in situ* benthic DOC fluxes could describe up to $\sim 28\%$ of $J_{DIC}$. At the same time, whether all the DOM utilisation that takes place within benthic chambers in our study is bound to the sediment-water interface is not clear. Thus, the enclosure of sediment for $\sim 30$ hrs may block out near bottom currents (e.g. Lüdke et al., 2019) and other mechanisms of lateral transport, e.g. eddies (Thomsen et al., 2016), that might influence the water column distribution of the freshly released from sediments DOM. For instance, Lüdke et al. (2019) reported near bottom poleward flow ranging from 0.1 to $0.4\,\mathrm{m\,s^{-1}}$. That could imply, that, at stable flow, DOM, which has been released by the sediment, could be distributed along with a distance of 10 to $40\,\mathrm{km}$ during the time equivalent to the time of sediment enclosure by BIGO chambers. Further-more, Loginova et al. (2016) reported an apparent transport of humic-like fluorescence to the surface waters. Therefore, DOM released to the bottom waters may not be limited only to the sediment-water interface, but instead may affect the whole water column biogeochemistry.

We suggest that the difference between the diffusive flux and net *in situ* flux could reflect the rate of microbial DOC utilisation at the sediment-water interface at each station. Thus, we estimated rates of microbial utilisation at St.3-St.6 ranging from 0.2 to $1.7\,\mathrm{mmol\,m^{-2}d^{-1}}$. We here propose to link these utilisation rates to rates of denitrification. Evidence from fieldwork suggests that at least part of the denitrification occurring at depth may be driven by the supply of POM via the biological carbon pump (Liu and Kaplan, 1984; Kalvelage et al., 2013). Others suggested that DOM supply could also stimulate denitrification in oxygen-deficient zones (e.g. Chang et al., 2014; Bonaglia et al., 2016). Given the importance of denitrification and N-loss rates for OMZ regions, it is crucial to evaluate all possible sources of organic matter potentially sustaining such rates. By conversion of the remineralisation rates of outfluxed DOM found in our study to denitrification rates using stoichiometry previously reported by Prokopenko et al. (2011), we estimated associated denitrification rates ranging from 0.2 to $1.4\,\mathrm{mmol\,m^{-2}d^{-1}}$. These are comparable to denitrification rates ($\sim 0.6\,\mathrm{mmol\,m^{-2}d^{-1}}$) and the total $N_2$ efflux ($\sim 1.2\,\mathrm{mmol\,m^{-2}d^{-1}}$) reported in anoxic sediments in the eastern tropical North Pacific off California (Prokopenko et al., 2011), to denitrification rates (0.2–2 $\mathrm{mmol\,m^{-2}d^{-1}}$) in the eastern tropical North Atlantic off Mauritania (Dale et al., 2014) and to modelled denitrification rates (0.5–1.1 $\mathrm{mmol\,m^{-2}d^{-1}}$) and $N_2$ fluxes (0.8–4.6 $\mathrm{mmol\,m^{-2}d^{-1}}$), observed along the 12°S transect (Dale et al., 2015; Sommer et al., 2016). Our estimates could, in turn, explain between 5 and $45\%$ of denitrification rates measured in the water column of the ETSP ($\sim 3\,\mathrm{mmol\,m^{-2}d^{-1}}$; Kalvelage et al., 2013). Therefore, we suggest that sediment release of DOC is not the dominant source of organic matter to the OMZ, but on occasions, this process may potentially serve as an important source of organic matter for the water column N–loss.

## 5 Conclusions

Diffusive fluxes of DOC and DON displayed high spatial variability, which was likely caused by the quality of organic matter supplied to the sediment and by differences in mechanisms of microbial metabolism at different water depths, suggested in the previous studies. Lower net *in situ* DOC and DON fluxes, compared to diffusive fluxes, as well as an apparent steepening

of $S_{275\text{-}295}$ and accumulation of humic-like material within benthic chambers during the time of the sediment enclosure at all stations, suggest that released to the water column DOM is actively reworked near the sediment. The remineralisation of DOM at the sediment-water interface is, likely, stimulated by high availability of strong electron acceptors, such as $NO_3^-$ and $NO_2^-$, in the water column at the outer shelf and continental slope stations. The utilisation of DOC released by the sediment, in turn,

may account for denitrification rates, comparable to previously reported for the water column and sediments off Peru and other OMZs (Kalvelage et al., 2013; Dale et al., 2014; Sommer et al., 2016), suggesting that sediment release, may serve as an important source of bioavailable DOM for the microbial communities at the sediment-water interface.

*Data availability.* All the measured DOC concentrations, $a_{CDOM}(325)$, $S_{275\text{-}295}$ and QSE of fluorescent components will be available at pangaea.de with the link to the project: SFB754 upon publication

*Author contributions.* ANL designed the sampling strategy and analysed DOM samples. AWD collected samples at MUC and BIGO stations and provided data for calculation of fluxes, ST helped with water sampling, DC provided inorganic N data, SS helped with the sampling strategy design and sampling and also provided all the facilities for sampling from BIGO landers, KW provided the initial idea for the research. ANL wrote the manuscript with contributions from AWD, FACLM, ST, SS, and AE.

*Competing interests.* The authors are not aware of competing interests of any sort for this research.

*Acknowledgements.* We are grateful to the chief scientists M. Dengler and ship and scientific crews of RV Meteor (cruises M136 and M137). J. Roa is acknowledged for DOC analyses. We are grateful to B. Domeyer, A. Bleyer, M. Türk and A. Beck for technical and logistical support, and U. Schroller-Lomnitz for advice. We thank T. Komada, P. Kowalczuk and an anonymous reviewer for improving this manuscript.

This research has been supported by the Deutsche Forschungsgemeinschaft (DFG) grant no.: SFB754 "Climate-Biogeochemical Interactions in the Tropical Ocean" (miniproposal, B9), the Helmholtz Association, and funding provided by the Inge-Lehmann-Fonds through

GEOMAR Helmholtz Centre for Ocean Research awarded to ANL; and by the funding provided by DFG Excellence cluster Future Ocean CP1403 "Transfer and remineralisation of biogenic elements in the tropical oxygen minimum zones" awarded to FACLM. ST was supported by the European Commission (Horizon 2020, MSCA-IF-2016, WACO 749699: Fine-scale Physics, Biogeochemistry and Climate Change in the West African Coastal Ocean).

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

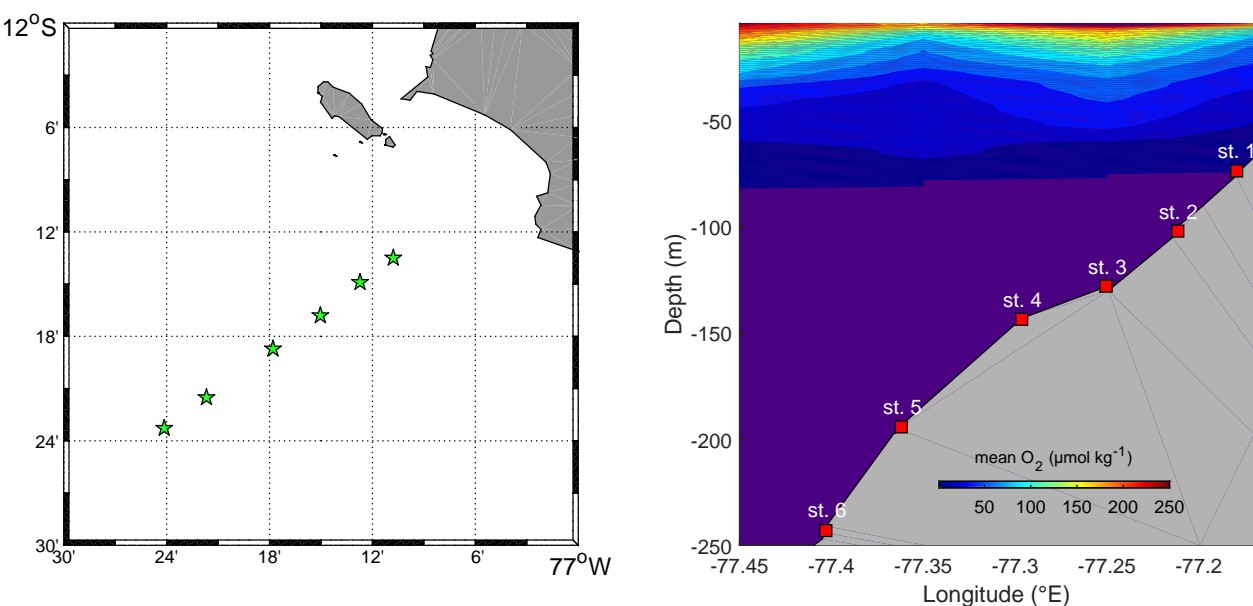

**Figure 1.** Left: distribution of sampling stations. Right: mean oxygen plot (the $O_2$ values were averaged over 1m depth and $0.1^o$ longitude intervals). The indigo colour represents values below 1 $\mu$mol kg$^{-1}$

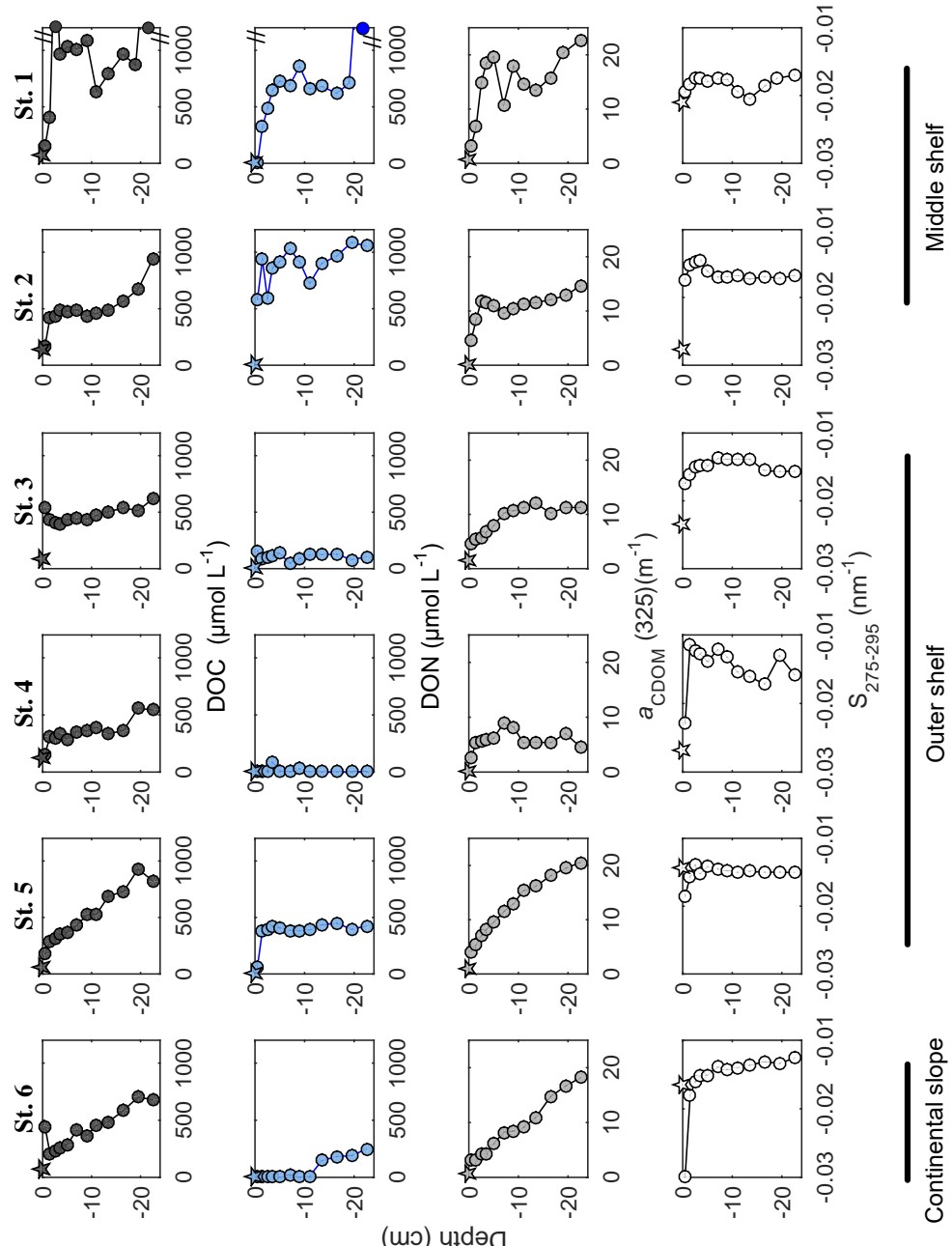

**Figure 2.** Porewater DOC (dark grey symbols), DON (blue symbols), $a_{CDOM}(325)$ (light grey symbols) and $S_{275-295}$ (white symbols) distribution within the sediment porewaters: depth profiles. Circles represent concentration/value, measured in the porewater sample, pentagrams represent the initial concentration/value within BIGO chambers. DOC concentrations after axis break are 2010 µmol L$^{-1}$ and 2568 µmol L$^{-1}$ at 3 cm and 22 cm sediment core depth, respectively. DON concentration after axis break is equal 2807 µmol L$^{-1}$

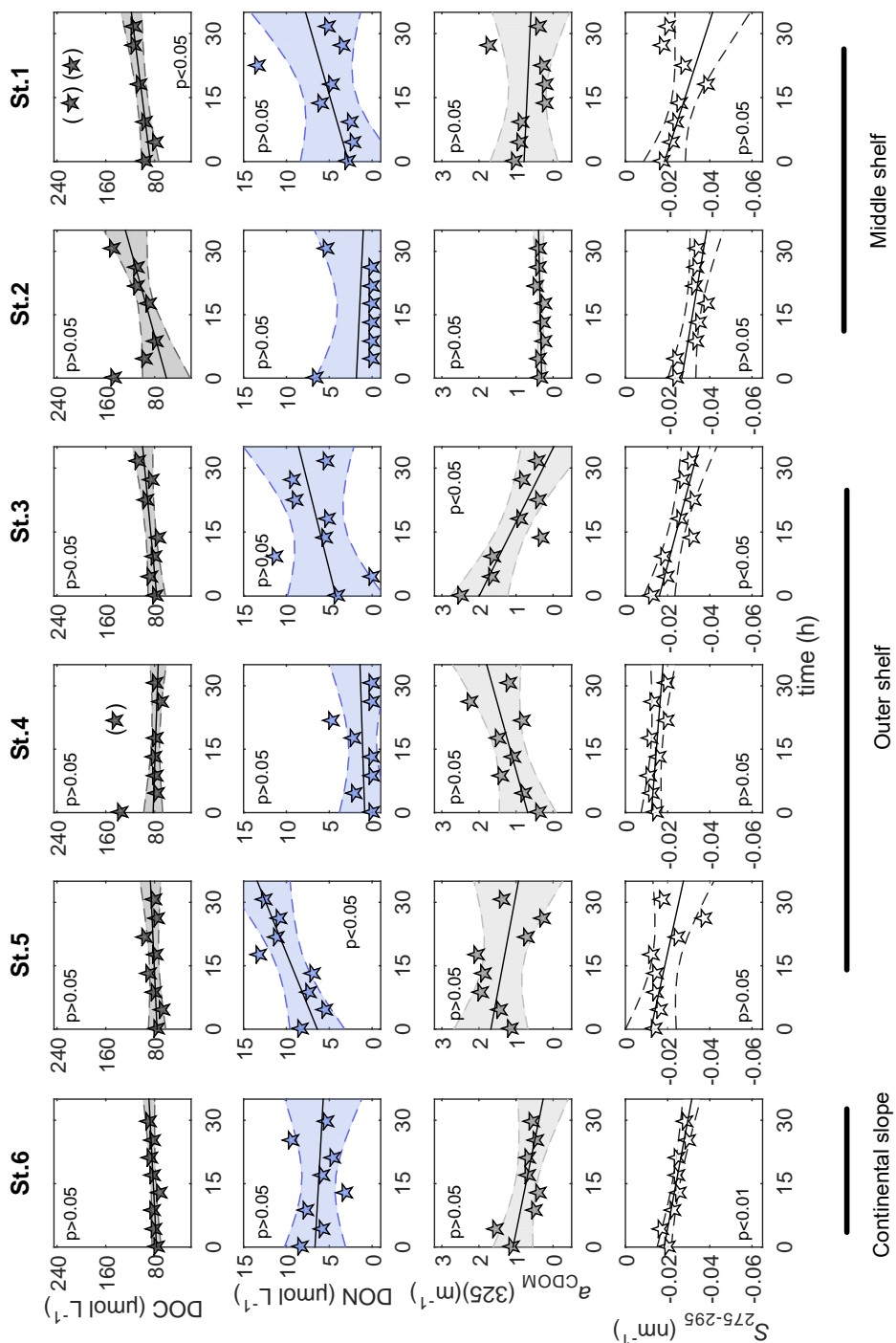

**Figure 3.** Distribution of DOC and CDOM parameters, $a_{CDOM}(325)$ and $S_{275-295}$, measured in BIGO chambers over time. Polynomial fit (1st order) was used for linear regression analyses: $t_0$ and data included in brackets were excluded from the analyses.

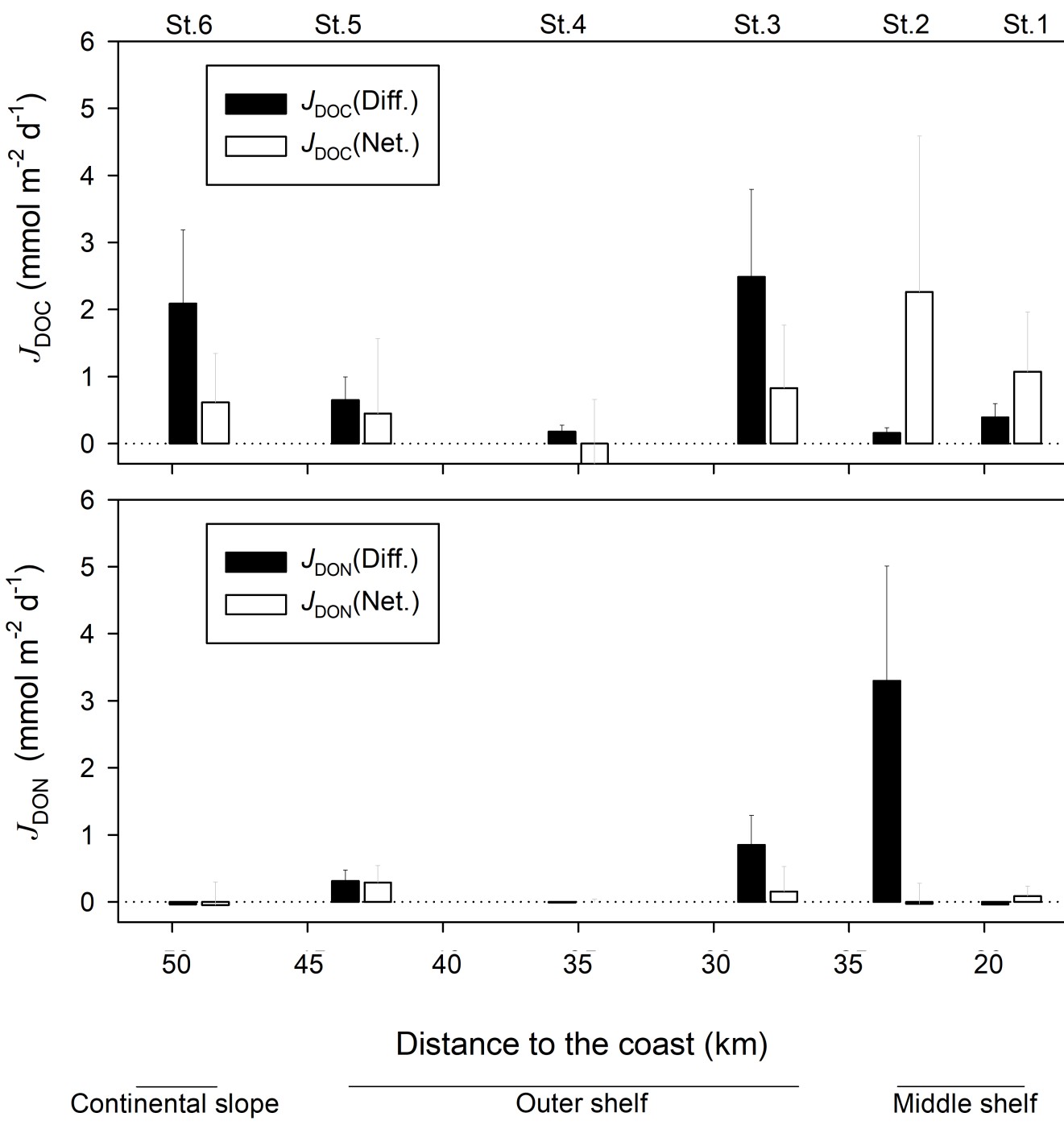

**Figure 4.** Diffusive and *in situ* net DOC (upper panel) and DON (lower panel) fluxes, evaluated at 12°S transect during this study.

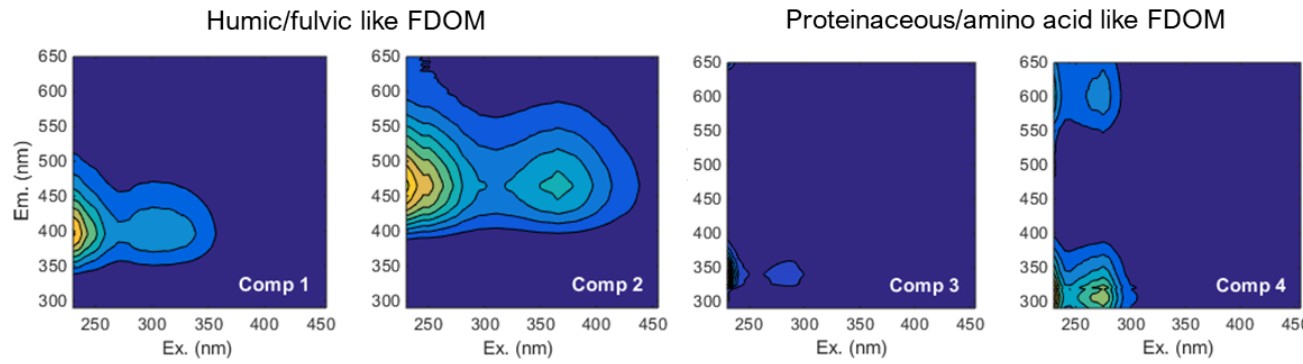

**Figure 5.** Four-components, which were found and validated by PARAFAC analyses after Murphy et al.(2013)

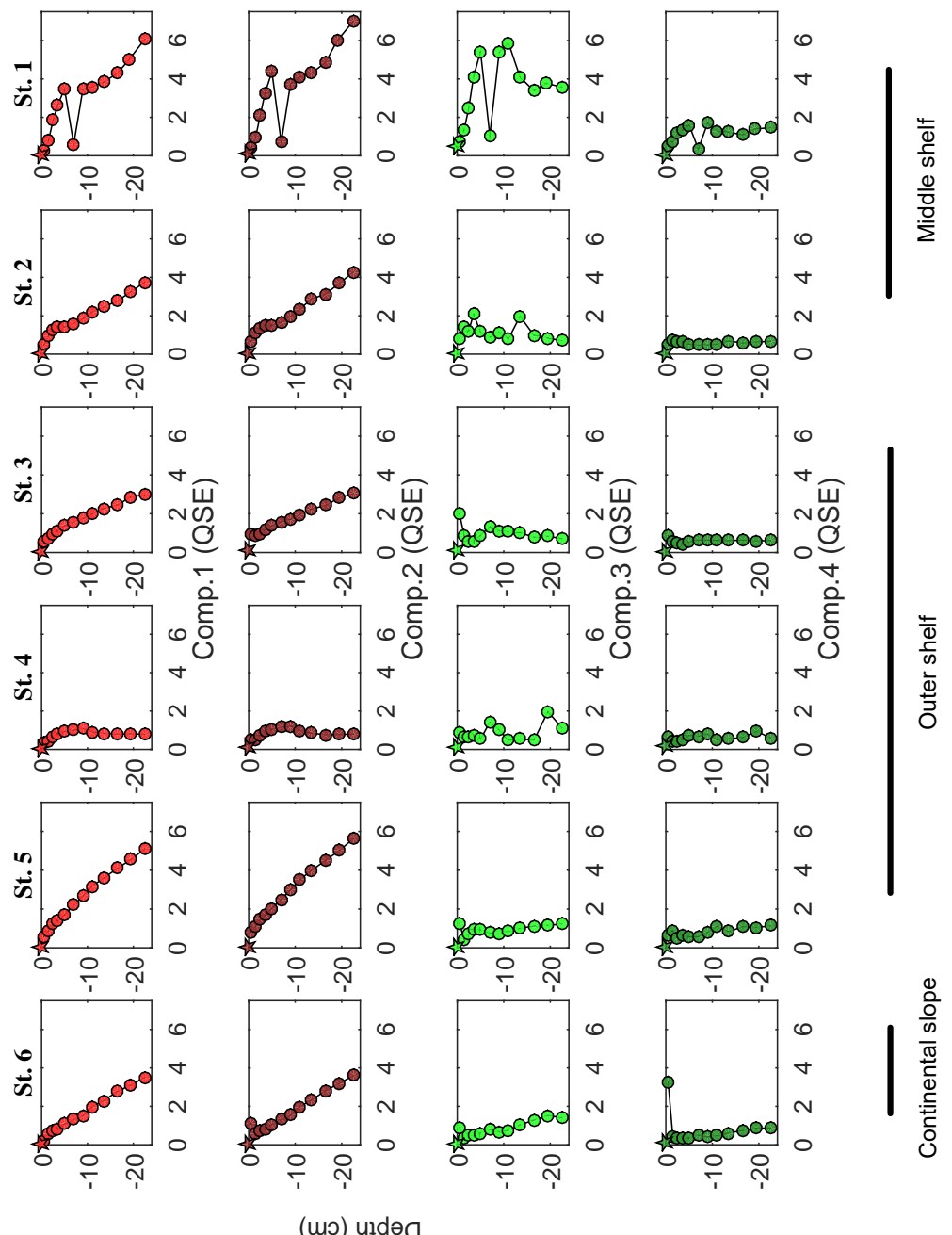

**Figure 6.** Porewater FDOM components distribution within the sediments: depth profiles. Humic-like Comp.1 and Comp.2 represented by light and dark red symbols, respectively. Amino acid-like Comp.3 and Comp.4 represented by light and dark green symbols, respectively. Circles represent concentration/value, measured in the porewater sample, pentagrams represent the initial concentration/value of the bottom water.

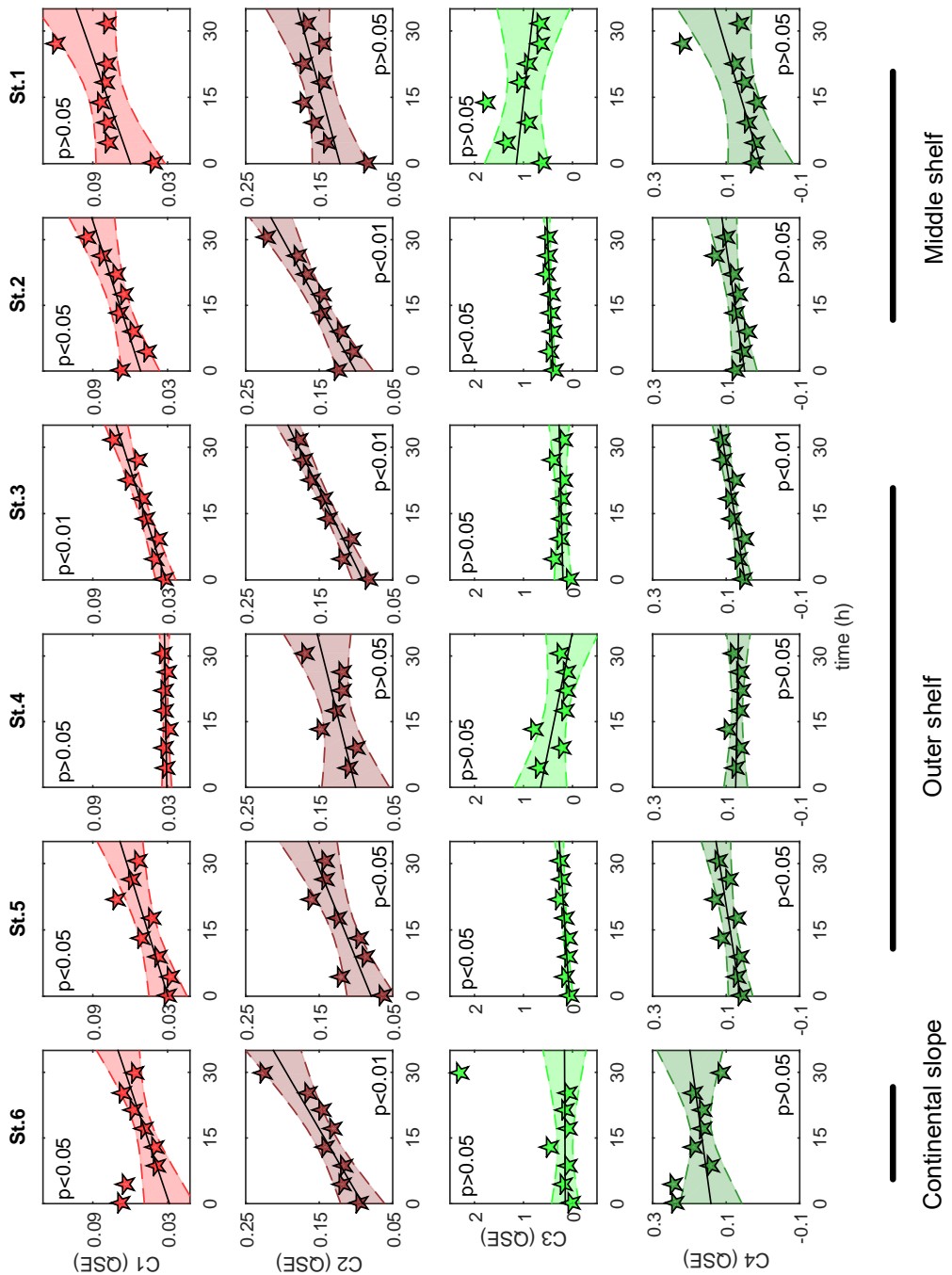

**Figure 7.** Distribution of FDOM components, measured in BIGO chambers over time. Polynomial fit (1st order) was used for linear regression analyses: $t_0$ values were excluded from the analyses.

**Table 1.** Stations and instruments deployed during our study on the Peruvian margin.

| Station | BIGO | MUC | Date (BIGO) | Date (MUC) | Latitude (°N) | Longitude (°E) | Depth (m) | Temp. (°C) | Porosity | $O_2$ (µmol kg$^{-1}$) | Dale et al. (2015) |
|---|---|---|---|---|---|---|---|---|---|---|---|
| St.1 | 533 BIGO II-IV | 483 MUC 8 | 27 Apr | 24 Apr | -77.180 | -12.225 | 74 | 16.2 | 0.93 | b.d. | Middle |
| St.2 | 642 BIGO II-II | 577 MUC 11 | 09 May | 01 May* | -77.212 | -12.248 | 102 | 15.9 | 0.96 | 11 | Shelf |
| St.3 | 488 BIGO II-III | 426 MUC 6 | 24 Apr | 19 Apr | -77.250 | -12.280 | 128 | 15.2 | 0.95 | b.d. | Outer |
| St.4 | 503 BIGO I-III | 651 MUC8 | 25 Apr | 10 May | -77.297 | -12.312 | 144 | 14.6 | 0.94 | b.d. | Shelf |
| St.5 | 471 BIGO I-II | 692 MUC 15 | 23 Apr | 13 May | -77.362 | -12.358 | 194 | 13.9 | 0.95 | b.d. | |
| St.6 | 415 BIGO II-I | 412 MUC 5 | 18 Apr | 18 Apr | -77.403 | -12.388 | 243 | 12.9 | 0.95 | b.d. | Continental Slope |

* $NH_4^+$ concentrations were measured at 787MUC33 on 20$^{th}$ of May at -12.247°N and -77.212°E.

Station depth was recorded from the ship winch. Bottom water temperature and $O_2$ are recorded by CTD. "b.d." stands for "below detection". Detection limit of $O_2$ is 5 µmol kg$^{-1}$ (Dale et al., 2015). Porosity is given for the upper 0.5 cm.