# Peer review of "Sediment release of dissolved organic matter to the oxygen minimum zone off Peru"

_Biogeosciences, 2019_

## Referee Comment (RC1) · Piotr Kowalczuk (Referee) · 28 Feb 2020

General opinion

Authors have presented a study on quantification of dissolved organic matter return flux and diffusion coefficient from bottom sediments to overlying waters in the tropical eastern South Pacific in the Peru upwelling zone. The Peru upwelling is one of five oceanic upwelling systems and is regarded one the most productive oceanic region globally. The primary production is sustained by the constant supply of inorganic nutrients to the euphotic zone, that simulate high growth rate of the autotrophic protists, that are base of the food web. The inorganic nutrients are product of the aerobic microbial processes leading to remineralization sinking particulate organic matter produced by

autotrophic protists in the euphotic zone. Aerobic microbial remineralization causes and decline of oxygen concentration in mesopelagic zone and below. Already published results of experiments and field studies in fjords and Baltic Sea coastal waters have provided evidence of the release of DOM from sediments to overlying water in anoxic conditions. Author have conducted their studies in the Peruvian upwelling system observing the differences in DOC, DON and CDOM and FDOM properties in the anoxic sediments in the continental shelf slope off Peru. They have quantified diffusion driven flux of DOC, CDOM and FDOM from sediments to near bottom waters. The overall DOC, and CDOM, DOM flux was low, and spatially variable. Using a chamber experiments Authors have found an accumulation of humic-like FDOM components in near bottom waters over time, that indicated active microbial reprocessing of FDOM released from sediments. The modification of DOM composition by microbial activity could be supported by high nitrate and nitrite concentration, and may lead to denitrification and loss of bio-available nitrogen in the near bottom waters.

The manuscript is very well written, and well edited serving as very important source if information on poorly described and quantified part of the DOM cycle. I found it very interesting and providing new and very relevant information. In my opinion this study deserved prompt publication in the current form.

Detailed remarks

Except of few typos error, that could be fixed during final edits I did not find any weak point in this presentation.

Piotr Kowalczuk

---

## Referee Comment (RC2) · Anonymous Referee #2 · 10 Mar 2020

Review of "Sediment release of dissolved organic matter to the oxygen minimum zone off Peru" by A.N. Loginova et al.

This manuscript reports assessments of benthic dissolved organic carbon (DOC) and dissolved organic nitrogen (DON) fluxes and pore water profiles from six sites on a transect of stations off central Peru. The chemical characteristics of DOM pools are also explored using absorbance and fluorescence spectral analyses. The work follows a series of other papers (e.g., Dale et al 2015 and 2016; Sommer et al 2016) reporting on benthic studies completed on research cruises to the Peruvian continental margin in 2017.

Generally, the manuscript was poorly prepared for external review. The English wording of sentences is often awkward, and many sentences contain extraneous words or are

missing key prepositions. Some of these problem sentences are listed below.

The paper presentation is also lacking depth and rigor. A more focused introduction and a much more informative description of the study area under section 2.1 are needed to set the stage for this work. The study area description should summarize the already published and spatially variable sediment carbon accumulation rates and benthic remineralization rates (e.g. DIC and nitrate fluxes) that are critical to the later discussion. This information could be incorporated into a more informative Figure 1. Meanwhile, Figure 2 is not needed and only repeats information given in the text about routine sampling and flux calculation methods.

With respect to the analytical work there are other concerns. There is no reporting of analytical blanks, precision or accuracy. I note the authors used cellulose acetate membrane syringe filters rather than combusted GF/F, so there could have been blank issues. The authors themselves raise the possibility that the DON results may be in error due to incomplete or unmatched estimates of total inorganic nitrogen species that must be subtracted from total dissolved nitrogen (TDN). Rather than speculate about this as they do near the bottom of page 9, have they any samples remaining to test for elevated NO3- stemming from either ammonia oxidation or bacterial sources? Any measurements of N2O? Were the samples completely processed under N2 to prevent oxidation artifacts? Can they report both TDN and inorganic N determinations (at least as supplemental material) so a reader can evaluate these together?

The presentation of flux determination approaches comes across as though the authors do not trust either the diffusive gradient approach or the results from in situ chambers (see for example the last two sentences on page 3). If it was my data set, I'd have greater confidence in the chamber-based fluxes, and I would view the fluxes calculated from the concentration difference across the sediment-water interface as "potential diffusive fluxes" that could result if there is no DOM source or sink at the sediment-water interface. Since most sites had mats of sulfide-oxidizing bacteria at the interface, microbial utilization as presented through Figure 9 seems likely and worthy of emphasis.

**BGD**

Differential diffusion rates and/or utilization rates of DOM pools are indicated by the FDOM components (Figure 8). These results are interesting, and they deserve more positive discussion.

The presentation of DOC and DON distributions and fluxes was uninspired. For some reason the authors simply compare mean $\pm$ sd of measurements, over whole profiles or incubations, across the stations. With all the available dissolved and solid phase biogeochemical data from these sites, they should look for relationships tied to organic matter degradation processes. For example, what do DIC or sulfate versus DOC, or ammonia versus DON property-property plots look like? There is much more that can be done to interpret these findings. The final speculative link to denitrification rates is completely unsupported.

Sentences with particularly awkward construction or in need of minor edits are found at:

Page 2: lines 12-15, line 34. Awkward.

Page 3 line 9, "or" not "of fulvic-like".

Page 3 line 13. Is the Uiam Lake study really relevant to a marine environment?

Page 3 line 18. Explain "insolation shield".

Page 3 line 22. Change to: "from pore-water gradients using. . ."

Page 3 line 27. Your point is unclear here. The uncertainty is in the sediment diffusion coefficient and whether DOM pools with different molecular weights are subject to different diffusion rates.

Page 4 lines 13-14. Unclear.

Page 9 lines 3-5. Awkward construction.

Page 9 line 35. Awkward.

Page 10 line 9. Change to "imbalance in production and consumption".

Page 10 line 15. Change to "agrees well with previous observations".

Page 10 line 24. Omit "to" before geopolymerization.

Page 12 line 18. Spelling "spatial".

Figure 7 caption, you use "stars" not pentagons.

Table 1. Units for dissolved oxygen are missing "micro" $\mu$.

―――――――――――――――――

---

## Referee Comment (RC3) · Tomoko Komada (Referee) · 22 Mar 2020

GENERAL COMMENTS: Benthic DOC and DON flux data are scarce, because they are difficult to obtain. The reactivity of the DOM that diffuse out of sediments is also not well constrained. This study is important in the sense that it contributes new data to both areas. However, as presented, I am not quite convinced that the conclusions drawn by the authors are fully supported by their findings.

SPECIFIC COMMENTS: Macrofauna are reported to be abundant in the study area. (In addition to what is discussed in the manuscript, Dale et al. (2015) mention occurrence of polychaetes at these stations, and Bohlen et al. (2011) report a bioturbation depth of 2 cm in the 11deg.S stations.) Centrifuging sediments containing macrofauna

has been shown to elevate DOC (Martin and McCorkle 1993, L&O, 38:1464-1479; Alperin et al. 1999, 63:427-448, GCA), and most probably DON. The authors should provide some evidence that assures the reader that their pore water DOM data are free of such artifacts. The authors report very low DOC/DON ratios in the sediment, and some spikes are apparent in DOC and DON in both depth profiles and in the chamber data. While microbial processes may be behind these features, it is also entirely plausible that they were due to occurrence of macrofauna (e.g., stirring up sediment during benthic chamber deployment; getting squashed in the centrifuge). This is a very important point to consider when comparing diffusive vs net (benthic chamber) DOM fluxes.

Syringe filters can give large DOC background (and possibly DON also), but there is no mention about how the filters were cleaned. Please provide additional information showing that the data do not contain high (and variable) levels of blank.

The authors state that microbial N turnover and DOM fluxes are likely related (page 9, line 11). I wholeheartedly agree with this statement, and find that this is an area that is ripe for further study. The authors go on to discuss N dynamics quite a bit, but the problem with this is that, other than DON, none of the inorganic N data are included in this manuscript. This renders most of the nitrogen-related discussion speculative at best. The authors should either scale back on the N discussion, include the DIN data, or perhaps plan on publishing a companion paper that includes relevant DIN data. At the very least, chamber data should include nitrate, assuming that was the major electron acceptor.

The DIN data are also relevant to the extremely low DOC/DON ratios in sediments. The authors originally declare that nitrate/nitrite concentrations in sediments were negligible (bottom of page 5), then resurrect this issue as a possible explanation for the low DOC/DON ratio (bottom of page 9), only to dismiss it again (top of page 10). The authors provide a few other possible explanations for the low DOC/DON ratios, but this discussion would be a lot more convincing in the presence of a more complete DIN

data showing that the DON values were not overestimated.

There seems to be an underlying assumption that sediment DOM is all refractory (e.g., page 1 line 5; page 2, line 21; page 11, line 15). As far as I am aware, this is not supported by the current literature. If anything, the opposite is more likely; a considerable fraction of DOM, especially near the sediment-water interface, is labile, and only a small fraction appears to be refractory (e.g., Bauer et al. 1995, Nature 373:686-689; Burdige et al. 2016, GCA 195:100-119; Komada et al. 2013, GCA 110:253-273). Therefore, assuming that the DOC and DON data presented here are indeed free of artifacts, it makes sense that the flux data point to microbial consumption of DOM that diffused out of the sediments. The authors should adjust the wording to better reflect the literature data.

TECHNICAL COMMENTS: I had difficulty reading Fig. 3, because the panels are so small. I am also unable to tell the difference between dark grey and blue (DOC vs DON). Black and grey arrows in Fig. 9 also look identical in color. Written English is OK, but not in publishable shape (parts that would benefit from editing are too numerous to list here). The narrative meanders in some places (e.g., discussion about DIN as I pointed out above). I also recommend streamlining the Introduction; I found the transition to DOC (line 16) a bit jarring.

―――――――――――――――――――――――

---

## Author Comment (AC1) · 5 May 2020

Replies to comments on the manuscript by Loginova et al. (BGD, 2020) In the following Reviewer's comments are marked as "R2" and the authors' responses are marked as "A".

R2: This manuscript reports assessments of benthic dissolved organic carbon (DOC) and dissolved organic nitrogen (DON) fluxes and pore water profiles from six sites on a transect of stations off central Peru. The chemical characteristics of DOM pools are also explored using absorbance and fluorescence spectral analyses. The work follows a series of other papers (e.g., Dale et al 2015 and 2016; Sommer et al 2016) reporting on benthic studies completed on research cruises to the Peruvian continental margin

in 2017. A: That is correct, and it also provides first measurements of dissolved organic matter in the pore waters and in benthic chambers in the area.

R2: Generally, the manuscript was poorly prepared for external review. The English wording of sentences is often awkward, and many sentences contain extraneous words or are missing key prepositions. Some of these problem sentences are listed below. A: The sentences listed below will be corrected according to the reviewer's suggestions in the reviewed manuscript. The reviewed version of manuscript will be checked by a native speaker.

R2: The paper presentation is also lacking depth and rigor. A more focused introduction and a much more informative description of the study area under section 2.1 are needed to set the stage for this work. The study area description should summarize the already published and spatially variable sediment carbon accumulation rates and benthic remineralization rates (e.g. DIC and nitrate fluxes) that are critical to the later discussion. This information could be incorporated into a more informative Figure 1. A: The Introduction will be restructured in the reviewed version of the manuscript into following:

[revised manuscript text omitted]

More information on previously published sediment type and fluxes will be added to the text of the revised manuscript on Page 4 line 25: "Sediments at the sampling stations are fine-grained diatomaceous dark-olive anoxic muds (Gutiérrez et al., 2009; Mosch et al., 2012) with porosity ranging between 0.8 and >0.9 (Table 1). Polychaetes and oligochaetes were found in the sampling area (Dale et al., 2015; Sommer et al., 2016). However, the sediment showed little evidence of strong mixing by bioturbation (Bohlen et al., 2011; Dale et al., 2015). In turn, the sediments are densely colonized by mats of large filamentous sulfur bacteria of the genera Tiloploca and Beggiatoa (Gutiérrez et al., 2009; Mosch et al., 2012). Dale et al. (2015) reported that mats of these sulphide oxidizing bacteria cover up to 100 % of the sediment surface at shallowest stations extending their trichomes 2 cm into the water column to access bottom water $NO_3-$. They could be observed from the sediment surface down to 20 cm sediment depth. At offshore stations, bacterial mats of several dm in diameter were covering up to 40 % of the sediment surface. Their occurrence was related to high carbon rain rates, which ranged from 10 mmol m-2 d-1 on the continental slope to 80 mmol m-2 d-1 on the shallowest shelf station (Fig. S1). Furthermore, the region was characterized by substantial organic matter utilization as indicated from high DIC fluxes and pore water $NH_4+$ concentrations (Dale et al., 2015). Thus, despite the highest sediment accumulation rates and POC content of the sediments, the highest organic matter respiration, as follows from large sediment DIC (Dale et al., 2015) and $NH_4+$ (Sommer et

al., 2016) fluxes at middle shelf stations, led to the smallest percentage of carbon burial efficiency (∼17%), compared to the outer shelf and the continental slope (24-74 %). Furthermore, Sommer et al. (2016) and Dale et al. (2016) suggested spatial variability of biological N cycling pathways in the area. Thus, outer shelf stations displayed the highest sediment uptake rate of NO3- and NO2- followed by high N2 outflux (Fig. S1). At shallower stations, NO3- and NO2- were entirely exhausted and excessively high fluxes of NH4+ were observed. Those spatial variabilities in N fluxes were suggested to be a result of dominating mechanisms of denitrification and anammox on the outer shelf and continental slope, and DNRA in the middle shelf. A further detailed description of the sediment and bottom waters at 12oS may be found in Dale et al. (2015, 2016) and Sommer et al. (2016)."

Instead of Figure 1 the discussed information from section 2.1 will be added to the Supplement as figure S1.

R2: Meanwhile, Figure 2 is not needed and only repeats information given in the text about routine sampling and flux calculation methods. A: Figure 2 will be moved to Supplement

R2: With respect to the analytical work there are other concerns. There is no reporting of analytical blanks, precision or accuracy. A: The following information will be added to the revised version of the manuscript:

to page5 lines 8-11:

"This method has a detection limit of ∼0.001 absorption units (that may be referred to ∼0.5 m-1) and a precision <5%, estimated as maximal standard deviation of CDOM absorbance spectra from 275 to 400 nm divided by the mean value of three repeated measurements.",

to page 5 lines 23-24:

"The precision of this method does not exceed 3% if estimated as a standard deviation

of Raman peaks at 275 nm of each measurement day, divided by the mean value.",

to page 5 line32 – Page6 line16:

"DOC samples were analysed by the high-temperature catalytic oxidation (TOC - VCSH, Shimadzu) with a detection limit of 1 $\mu$mol L-1 as described in detail by Engel and Galgani (2016). Calibration of the instrument was performed every second week using six standard solutions of 0, 500, 1000, 1500, 2500 and 5000 $\mu$g C L-1, which were prepared using a potassium hydrogen phthalate standard (Merck 109017). Before each set of measurements, a baseline of the instrument was set using ultrapure water. The deep-sea standard (Dennis Hansell, RSMAS, University of Miami) with known DOC concentration was measured after setting the baseline to verify accuracy by the instrument. Typically, the precision of the method did not exceed 4 %. Furthermore, two control samples with known concentrations of DOC were prepared for each day of measurement using a potassium hydrogen phthalate standard (Merck 109017). The DOC concentrations of those control samples were typically within the range of samples' concentrations and were measured at the time of sample analyses to control baseline flow during measurements. The DOC concentration was determined in each sample out of five to eight replicate injections. A TNM-1 N detector of Shimadzu analyser was used to determine total dissolved nitrogen (TDN) in parallel to DOC with a detection limit of 2 $\mu$mol L-1 (Dickson et al., 2007). Calibration was performed simultaneously with the calibration of carbon detector using standard solutions of 0, 100, 250, 500 and 800 $\mu$g N L-1, which was prepared using potassium nitrate Suprapur (Merck 105065). The deep-sea standard (Dennis Hansell, RSMAS, University of Miami) with the known concentration of TDN was measured daily to verify the accuracy of the instrument. The precision of the method did not exceed 2 % estimated as the standard deviation of 5–8 injections divided by the mean value. Concentrations of DON were calculated as a difference of TDN and the sum of concentrations of inorganic N components."

R2: I note the authors used cellulose acetate membrane syringe filters rather than

combusted GF/F, so there could have been blank issues. A: We thank the Reviewer 2 for noticing that. Indeed, we did not add the details behind choosing a filter type. As collected samples were expected to be highly concentrated. Due to relatively long storage of our unfixed CDOM and FDOM samples prior to analyses, we thought of the easy way for removing most of the bacteria. Therefore, using the pore size of 0.2 $\mu$m rather than 0.7 $\mu$m (as GFF may give) was preferred. Prior to the research cruise, we did several checks for different filters of that pore size, which are commonly used during pore water work, including PES, nylon, CA and RC. All the filters gave one or another background level, therefore, we tested which volume of ultrapure water was the optimal for cleaning. CA and RC filters gave the minimum values for DOC and for DON after rinsing with 60 ml of ultrapure water. CA was chosen over RC due to lower binding affinity to macromolecules and proteins, as we did not want to influence recovery of organic components during filtration.

Following will be added to the page 4 lines 25-30 to the revised version of the manuscript: "All samples were passed through pre-washed (60 mL of ultrapure water) cellulose acetate (CA) membrane syringe filters (0.2 $\mu$m) and first five mL of the filtrate were discarded to waste before filling the sample into storage vials. Several types of filters (PES, nylon, CA and regenerated cellulose (RC)) were examined for background DOC and total dissolved nitrogen (TDN) signal before the cruise. CA and RC filters gave minimal background concentrations for both parameters after rinsing with 60 ml of ultrapure water (Fig.S4). CA filters were chosen over RC due to their lower binding affinity to macromolecules and proteins." Figure S4 will be also added to the Supplement.

R2: The authors themselves raise the possibility that the DON results may be in error due to incomplete or unmatched estimates of total inorganic nitrogen species that must be subtracted from total dissolved nitrogen (TDN). Rather than speculate about this as they do near the bottom of page 9, have they any samples remaining to test for elevated $NO_3^-$ stemming from either ammonia oxidation or bacterial sources? Any

Interactive
comment

measurements of N2O? A: Unfortunately, we do not have spare samples left. We will omit the speculative discussion from the chapter 4.1 of the revised manuscript: "For instance, NO3 that is present at high concentrations in intracellular vacuoles of Marthioploca (Dale et al., 2016) could be leaked to the pore water during sediment handling and centrifugation. An ammonium oxidizing bacteria were shown previously to be able profiting from nitrous oxide, produced by denitrification (e.g. Kartal et al., 2013). Thus, the production of NH+4, as a result of DNRA occurring at the inner shelf stations in combination to nitrous oxide production via denitrification occurring at outer shelf, may produce a convenient niche for anammox bacteria at the rim of the inner shelf at 12oS. The intermediate product of anammox, hydrazine (e.g. Kartal et al., 2013), may, in turn, accumulate in the inner space of anammox bacteria, and be released in the pore water samples as a consequence of the cell rapture induced by centrifugation. However, the concentrations of those intermediate products are likely very small and may not explain elevated TDN values."

R2: Were the samples completely processed under N2 to prevent oxidation artifacts? A: That is correct, the sediment cores were processed under N2 atmosphere up to the point of filtering the centrifuged samples and then adding acid to the DOC samples. The sealing with fire was not possible inside the glove bag. CDOM and FDOM samples were filled under air, however, we would not expect immediate changes in optical properties. Furthermore, CDOM and FDOM samples are stored in tightly sealed vials although not under anoxic atmosphere, as this makes the transport to home laboratory very challenging. We will add following to the chapter 2.2: "Retrieved sediments were immediately transferred to the onboard cool room (10-15 C°) and processed under anoxic conditions within few hours using an argon-filled glove bag."

R2: Can they report both TDN and inorganic N determinations (at least as supplemental material) so a reader can evaluate these together? A: Unfortunately, we were restricted by the data legacy and could not report on DIN from benthic chambers and pore waters, but could only use the data for our calculations of DON. The data on DIN

from the benthic chambers will be published soon in a different manuscript by MSc David Clements and co-authors. However, MSc Clements has agreed to provide us the data for DIN for publishing from one of the stations. Therefore, data for DIN components from one benthic chamber at station 3 will be added to a Supplement as a Figure S6. Due to reviewers' suggestions, measurements of ammonia may now be published for all six stations and will be added to a Supplement as a depth profile plot (Figure S5).

R2: The presentation of flux determination approaches comes across as though the authors do not trust either the diffusive gradient approach or the results from in situ chambers (see for example the last two sentences on page 3). If it was my data set, I'd have greater confidence in the chamber-based fluxes, and I would view the fluxes calculated from the concentration difference across the sediment-water interface as "potential diffusive fluxes" that could result if there is no DOM source or sink at the sediment-water interface. A: We understand the reviewer's concern. The text:" The release of dissolved substances from anoxic sediments is regulated mainly by diffusion through the sediment–water interface (e.g. Lavery et al., 2001, and references therein). Diffusion–driven solute fluxes (hereon "diffusive fluxes") are commonly evaluated from pore-waters gradient using Fick's First Law. Diffusive DOM fluxes have been found to be consistent with total DOM flux in non-bioturbated anoxic sediments (Burdige et al., 1992), such as those found off Peru (Dale et al., 2015; Sommer et al., 2016). In some sediments, however, the diffusive flux may overestimate the total flux (Burdige et al., 1992; Lavery et al., 2001). This may be attributed to bioturbation, "unfavourable" redox conditions (Lavery et al., 2001), irreversible adsorption onto particles, and biological DOM consumption at the sediment–water interface or in the bottom waters (Burdige et al., 1992). Furthermore, the assumptions or calculations of certain DOM parameters, such as molecular weight (Balch and Guéguen, 2015) and tortuosity (Ullman and Aller, 1982) may induce potential bias to the flux calculations. In situ measurements of the net solute flux using benthic incubation chambers are independent from molecular weight and tortuosity uncertainties. This approach is laborious

and based on the assumption that solutes, released into the benthic chamber, behave conservatively during the time incubation, and, show linear trends over time. Herewith, the in-situ measurements may be affected by an accidental enclosure of benthic macro-organisms, such as for instance Pleuroncodes mondon, which are abundant in the Peruvian OMZ (Kiko et al., 2015)." -on page 3, lines 20-35 will be rephrased to:

"The release of dissolved substances from anoxic sediments is regulated mainly by diffusion through the sediment—water interface (e.g. Lavery et al., 2001, and references therein). Diffusion–driven DOM fluxes (hereon "diffusive fluxes") and net DOM fluxes (hereon "net fluxes") are commonly evaluated from pore-water gradients using Fick's First Law and by enclosing and incubating a small area of the sediment surface over time, respectively. Diffusive DOM fluxes have been found to be consistent with net DOM flux in non-bioturbated anoxic sediments (Burdige et al., 1992). In some sediments, however, the diffusive flux may overestimate the net flux (Burdige et al., 1992; Lavery et al., 2001). This may be attributed to bioturbation, "unfavourable" redox conditions (Lavery et al., 2001), irreversible adsorption onto particles, and biological DOM consumption at the sediment—water interface or in the bottom waters (Burdige et al., 1992). The determination of in situ net DOM fluxes using benthic incubation chambers are independent of such uncertainties. This approach bases on the assumption that solutes, released into the benthic chamber, behave conservatively during the time incubation, and, show linear trends over time." As it goes through discussion, we suggest that diffusive fluxes are consumed at the surface sediment-bottom water interface in agreement with the reviewer's remarks.

R2: Since most sites had mats of sulfide-oxidizing bacteria at the interface, microbial utilization as presented through Figure 9 seems likely and worthy of emphasis. A: We are not very sure how to understand/ implement this comment to the revised manuscript, as our paper, and Figure 9 in particular, was supposed to reflect that a part of the DOM sediment release, that may be provided through diffusion, is utilized by microbial communities, resulting in the flux, that was obtained in benthic chambers.

R2: Differential diffusion rates and/or utilization rates of DOM pools are indicated by the FDOM components (Figure 8). These results are interesting, and they deserve more positive discussion. A: We appreciate that the Reviewer 2 gives a value to our optical data. The following will be added to the reviewed version of the manuscript at:

Page 10, line 25:

Accordingly, pore water DOM optical properties reflected the "freshest" character of organic matter at St.1 and St.2, whereby S275-295 displayed similar properties to those in the water column (Fig. 3), an enrichment in protein-like DOM fluorescence (Fig. 6) and in DON (Fig. 3). Therefore, in line to the previous findings, our data suggests that the middle shelf stations are supplied with more labile POM compared to the outer shelf stations. This labile POM, likely of proteinaceous origin (e.g. Faganeli and Herndl, 1991), is rapidly reworked, resulting in greater DOM release at the middle shelf stations. However, despite the highest sediment accumulation and POC mineralization rates at St.1 (Dale et al., 2015) and the "freshest" DOM character, the diffusive fluxes of DOC and DON here were not highest on the transect even though pore waters showed elevated DOM levels (Fig. 8). As aCDOM(325) and protein-like FDOM was previously related to the dynamics of labile DOM (Loginova et al., 2016), one may expect those fractions to be rapidly reworked by heterotrophic communities. Therefore, little dynamics of optical properties of proteinaceous character and aCDOM(325) might be a result of not only of the absence of benthic labile DOM fluxes, but also a signature of rapid microbial utilization of labile organic matter freshly released from the sediment (Komada et al., 2016). Thus, the greatest decrease in S275-295 and accumulation of humic-like substances suggest that benthic release of fresh bioavailable DOM should be rapidly and extensively reworked or consumed at the sediment—water column interface during the time of incubations at St.1. In turn, these results support the idea that microbial utilization is controlled by the quality of supplied organic matter (Pantoja et al., 2009; Le Moigne et al., 2017)."

Page 11, line 14:

"In agreement to this S275-295 revealed lowest changes over time, suggesting that DOM at benthic chamber at St.2 remains "fresh" during the time of incubations. Similar to that proteinaceous Comp.3, despite its generally low variability, exhibit highest increase at St.2, suggesting relative accumulation of proteinaceous DOM in the corresponding chamber."

R2: The presentation of DOC and DON distributions and fluxes was uninspired. For some reason the authors simply compare mean $\pm$ sd of measurements, over whole profiles or incubations, across the stations. A: DOC and DON results will be rephrased in the reviewed version of the manuscript Page: 8 Line: 16 into: "Pore-water DOC generally accumulated with depth in the sediment (Fig.2). Highest concentrations of DOC were measured at the middle shelf at station 1 (St.1), ranging from 152 $\mu$mol L-1 at 0.5 cm to a maximum of 2.6 x103 $\mu$mol L-1 at 22.5 cm of sediment depth. Pore-water DOC concentrations and gradients decreased gradually towards station 4 (St.4), where DOC concentrations ranged from 122 $\mu$mol L-1 at 0.5 cm to 544 $\mu$mol L-1 at 22.5 cm of sediment depth. Further offshore, pore water DOC concentrations and gradients increased at station 5 (St.5) and station 6 (St.6), ranging from 177 $\mu$mol L-1 at 0.5 cm to 823 $\mu$mol L-1 at 22.5 cm and from 210 $\mu$mol L-1 at 1.5 cm to 702 $\mu$mol L-1 at 19.5 cm, respectively. Porewater DON was largely influenced by vicinity to the coast (Fig. S7). Highest concentrations of DON were measured at the middle shelf St.1 and St.2. The DON concentrations in pore waters at these stations were ranging from b.d.l. at 0.5 cm to a maximum of 2.6 x103 $\mu$mol L-1 at 22.5 cm and from 580 $\mu$mol L-1 at 0.5 cm to 1.1 x103 $\mu$mol L-1 at 19.5 cm of sediment depth, respectively. Similarly to DOC, the pore water DON concentrations decreased towards St.4, where they ranged from b.d.l. at surface sediment to 85 $\mu$mol L-1 at 3.5 cm sediment depth and then resumed the gradient offshore at St.5 (64—450 $\mu$mol L-1) and St.6 (b.d.l.—248 $\mu$mol L-1)."

R2: With all the available dissolved and solid phase biogeochemical data from these sites, they should look for relationships tied to organic matter degradation processes. For example, what do DIC or sulfate versus DOC, or ammonia versus DON propertyproperty plots look like? There is much more that can be done to interpret these findings. A: We very much understand the wish of the Reviewer R2 for a more extended data analysis to better constrain the link between the degradation of organic matter in the sediment and resulting DOC fluxes. However, as mentioned previously the fluxes of DIC and ammonium measured during the same cruises (M136/M137) are essential part of an ongoing PhD thesis and are not yet published. Hence, please understand that we are hesitating in providing these data in this manuscript which might endanger the originality of the PhD Thesis. We explored possible links between organic carbon degradation and DOC/DON fluxes using data from a previous cruise, but became aware that the different bottom water concentrations during these different cruises might introduce more uncertainties into the data interpretation. We consider the present manuscript as one of the first studies addressing DOC and DON fluxes measured in situ using benthic landers. A more deeply rooted synthesis paper will become possible when data from all cruises made in the SFB754 becomes available.

R2: The final speculative link to denitrification rates is completely unsupported. A: In the final paragraph of the discussion, we proposed to link estimated rates of DOC supply to that of denitrification processes. Denitrification processes are not uncommon in regions where O2 concentration is low (Lam and Kuypers, 2011). Evidences from various fieldwork suggest that at least part of the denitrification occurring at depths may be driven by the supply of OM (Liu and Kaplan 1983, Kalvelage et al., 2013). Some of these work proposed that the biological carbon pump (POC downward export) as one potential supply pathways of OM sustaining deep water denitrification (Kalvelage et al., 2013). Other suggested that DOM supply could also stimulate denitrification in anoxic waters (e.g. Chang et al., 2014, Bonaglia et al., 2016). Given the importance of denitrification and N loss for OMZ regions, it is crucial to constrain potential sources of OM potentially sustaining such rates. We show that the supply of OM from sediment release (and subsequent remineralisation) can be large. Such releases can be transported, remineralised Prokopenko et al. (2011) and potentially used by denitrifiers in the water column. It is therefore not irrelevant to provide numbers on the amount of N loss driven

by DOC sediment release may these be upper bound estimates. We simply aimed here to confront potential DOC-sediment releases derived denitrification rates to that of BCP derived rates (provided in (Kalvelage et al., 2013)). In essence, we supported our estimation by stoichiometrically converting sediment DOC release respiration rates into denitrification rates using stoichiometry previously reported by Prokopenko et al. (2011). Our estimations are within the range what is usually observed and estimated for similar regions. This further supports our approach. We state our statement by providing upper and lower range DOC-sediment releases derived denitrification rates based on the upper and lower measurement of DOC-sediment releases turnover rates (See Figure 8). Our DOC-sediment releases derived denitrification rates range now from 0.2 to 1.4 mmol m−2d−1. This in turn could explain between 5 and 45 % of denitrification rates measured in the water column in the eastern tropical South Pacific (âĹij3mmol m−2d−1; Kalvelage et al., 2013). This suggests that on occasion, sediment release of DOC may potentially serve as an important organic matter source for the water column N–loss as originally stated. We have modified the text in the revised version of the manuscript to better explain the relevance as well as the uncertainties associated to our approach (providing lower and upper bound on proportion of denitrification potentially explained by DOC sediment releases). We hope that this will satisfy the reviewer.

The text: "We suggest that the difference between the diffusive flux and netin situflux could reflect the rate of microbial DOC uti-15lization in the chamber water and/or surface sediment layer at each station. Thus, the rate of the microbial utilization at St.3–St.6 ranged from 0.2 to 1.7mmol m−2d−1(Fig. 8). These consumption rates could support a denitrification rate of 0.2–1.4mmol m−2d−1, based on reaction stoichiometry reported by Prokopenko et al. (2011). These are comparable to denitrification (0.6±0.4mmol m−2d−1) and the total N2efflux (âĹij1.2mmol m−2d−1) in anoxic sediments in the eastern tropical NorthPacific off California (Prokopenko et al., 2011), to denitrification rates (0.2–2mmol m−2d−1) in the eastern tropical North At-20lantic off Mauritania (Dale et al., 2014) and to modelled denitrification rates (0.5–1.1mmol

m−2d−1) and N2fluxes (0.8–4.6mmol m−2d−1), observed along 12oS transect (Dale et al., 2016; Sommer et al., 2016). Furthermore, the estimated potentialdenitrification rates may be able to explain up toâĹij55%of denitrification rates in the water column in the eastern tropicalSouth Pacific (âĹij3mmol m−2d−1Kalvelage et al., 2013), suggesting that sediment release of DOM may potentially serve asan important organic matter source for the water column N–loss."

will be changed to: "... Therefore, DOM released to the bottom waters may be not limited only to the sediment—water column interface, affecting whole water column biogeochemistry. We suggest that the difference between the diffusive flux and net in situ flux could reflect the rate of microbial DOC utilization in the chamber water and/or surface sediment layer at each station. Thus, we estimate rates of microbial utilization at St.3–St.6 ranging from 0.2 to 1.7mmol m−2d−1(Fig. 8). We here propose to link these to that of denitrification processes. Evidences from fieldwork suggest that at least part of the denitrification occurring at depth may be driven by the supply of POM via the biological carbon pump (Kalvelage et al., 2013). Other suggested that DOM supply could also stimulate denitrification in oxygen deficient zones (e.g. Chang et al., 2014, Bonaglia et al., 2016). Given the importance of denitrification and N-loss rates for OMZ regions, it is crucial to evaluate various possible sources of OM potentially sustaining such rates. Conversion of the remineralisation rates of outfluxed DOM, found in our study (Fig. 8), into denitrification rates using stoichiometry previously reported by Prokopenko et al. (2011), we estimate associated denitrification rates ranging from 0.2 to 1.4 mmol m−2d−1. These are comparable to denitrification rates (0.6±0.4 mmol m−2d−1) and the total N2 efflux (âĹij1.2 mmol m−2d−1) reported in anoxic sediments in the eastern tropical North Pacific off California (Prokopenko et al., 2011), to denitrification rates (0.2–2mmol m−2d−1) in the eastern tropical North Atlantic off Mauritania (Dale et al., 2014) and to modelled denitrification rates (0.5–1.1 mmol m−2d−1) and N2 fluxes (0.8–4.6 mmol m−2d−1), observed along 12°S transect (Dale et al., 2016; Sommer et al., 2016). Our estimates could, in turn, explain between 5 and 45 % of denitrification rates measured in the water column in the eastern tropical South Pacific

(âĹij3 mmol m−2d−1; Kalvelage et al., 2013). We suggest that sediment release of DOC is not the dominant source of OM to the OMZ, but on occasions, this process may potentially serve as an important source of organic matter source for the water column N–loss."

R2: Sentences with particularly awkward construction or in need of minor edits are found at: R2: Page2 lines 12-15: Awkward A: "Extensive fieldwork campaigns conducted on anoxic Peruvian sediments suggested further show that they act as "factories" for an intensive organic matter remineralization (Dale et al., 2015). Yet, the burial efficiency of particulate organic carbon (POC) varies throughout OMZ (Dale et al., 2015). For instance, burial efficiency are low at anoxic inner shelf stations despite highest carbon mineralization rates estimated from in situ dissolved inorganic carbon (DIC) fluxes (Dale et al., 2015)."

will be changed to: "Similar to the water column studies, extensive fieldwork campaigns conducted on sediments off Peru also suggested intensive particulate organic matter (POM) remineralization under full anoxia (Dale et al., 2015).

While POM degradation in sediments is mostly associated with its full remineralization to dissolved inorganic carbon (DIC) and inorganic nutrients, the mechanism of POM remineralisation implies important intermediate stages of dissolved organic matter (DOM) production, reworking and mineralization processes (Smith et al., 1992; Komada et al., 2013). Thus, around 10 % of remineralized particulate organic carbon (POC) may accumulate as dissolved organic carbon (DOC) in the pore waters (Alperin et al., 1999). In turn, DOM efflux may represent an important escape mechanism for carbon from sediments (e.g. Ludwig et al., 1996; Burdige et al., 1999) and a source of organic matter to the water column (e.g. Burdige et al., 2016). Despite the acknowledged importance of sediment DOM for organic matter cycling, the measurements of benthic DOM fluxes remain scarce and the reactivity of the pore-water DOM is not well constrained."

[Figure]

R2: Page2 lines 34: Awkward A: "CDOM absorbance spectra represent an exponential curve with no discernible peaks" will be changed to: "Typical CDOM absorbance spectrum is shaped as an exponential curve"

R2: Page3 lines 18: explain "insolation shield" A: "insolation shield" will be changed to "reduce penetration of hazardous or bioavailable light"

R2: Page3 lines 22: change to "from pore water gradients using" A: will be changed

R2: Page3 lines 27: Your point is unclear here. The uncertainty is in the sediment diffusion coefficient and whether DOM pools with different molecular weights are subject to different diffusion rates. A: Following will be removed from revised version: "Furthermore, the assumptions or calculations of certain DOM parameters, such as molecular weight (Balch and Guéguen, 2015) and tortuosity (Ullman and Aller, 1982) may induce potential bias to the flux calculations."

R2: Page4 lines 13-14: unclear A: As the reviewer suggested more informative description of the study area, the brief description of the study area: "A detailed description of the sediment at 12oS is reported in Dale et al. (2015, 2016). In brief, sediments at the sampling stations are fine-grained muds with porosity ranging between 0.8 and 0.95 (Dale et al., 2015; Sommer et al., 2016)" will be omitted from the edited manuscript.

R2: Page9 lines 3-5: Awkward construction A: "The data suggests that the inner shelf stations receive of the most labile POM, likely of proteinaceous origin (e.g. Faganeli and Herndl, 1991) compared to the outer shelf stations, which is likely being rapidly reworked into DOM at the inner shelf compared to the other sites." Will be changed to: "Our data suggests that the inner shelf stations receive of the most labile POM compared to the outer shelf stations. This labile POM, likely of proteinaceous origin (e.g. Faganeli and Herndl, 1991), is rapidly reworked, resulting in greater DOM release in the inner shelf stations.

R2: Page9 lines 35: Awkward A: "An ammonium oxidizing bacteria were shown previously to be able profiting from nitrous oxide, produced by denitrification (e.g. Kartal et al., 2013)." Will be changed to: "As ammonium oxidizing bacteria profit from nitrous oxide, produced by denitrification (e.g. Kartal et al., 2013)."

R2: Page 10 line 9. Change to "imbalance in production and consumption". A: will be changed to "imbalance in production and consumption"

R2: Page 10 line 15. Change to "agrees well with previous observations". A: will be changed to "agrees well with previous observations"

R2: Page 10 line 24. Omit "to" before geopolymerization. A: "to" will be deleted

R2: Page 12 line 18. Spelling "spatial". A: will be corrected

R2: Figure 7 caption, you use "stars" not pentagons. A: "pentagons" will be changed to "pentagrams" in the plots' descriptions.

R2: Table 1. Units for dissolved oxygen are missing "micro" $\mu$ A: $\mu$ will be added.

---

## Author Comment (AC2) · 5 May 2020

In the following, comments by Dr. Tomoko Komada are marked as "TK" and authors' responses are marked as "A".

TK: Benthic DOC and DON flux data are scarce, because they are difficult to obtain. The reactivity of the DOM that diffuse out of sediments is also not well constrained. This study is important in the sense that it contributes new data to both areas. However, as presented, I am not quite convinced that the conclusions drawn by the authors are fully supported by their findings. SPECIFIC COMMENTS: TK: Macrofauna are reported to be abundant in the study area. (In addition to what is discussed in the manuscript, Dale et al. (2015) mention occurrence of polychaetes at these stations, and Bohlen et al.

(2011) report a bioturbation depth of 2 cm in the 11deg.S stations.) Centrifuging sediments containing macrofauna has been shown to elevate DOC (Martin and McCorkle 1993, L&O, 38:1464-1479; Alperin et al. 1999, 63:427-448, GCA), and most probably DON. The authors should provide some evidence that assures the reader that their pore water DOM data are free of such artifacts. The authors report very low DOC/DON ratios in the sediment, and some spikes are apparent in DOC and DON in both depth profiles and in the chamber data. While microbial processes may be behind these features, it is also entirely plausible that they were due to occurrence of macrofauna (e.g., stirring up sediment during benthic chamber deployment; getting squashed in the centrifuge). This is a very important point to consider when comparing diffusive vs net (benthic chamber) DOM fluxes. A: This is a good point, and we thank Dr. Komada for mentioning that. We had no control over this question; however, we may refer to logistical reasons and previous studies in order to explain our choice of the centrifugation as a method for DOM extraction. First of all, we have chosen centrifugation over direct squeezing, as the latter method would imply numerous soft plastic parts, that were not possible to be pre-cleaned in advance. The centrifugation tubes (PP), in turn, were pre-cleaned with HCl, for each sample individual PP tube could be used. We will add the following to the methods section: "Studies conducted in areas with abundant macrofauna suggested that pore waters isolated by centrifugation exhibit higher DOC concentrations compared to for non-invasive methods, such as sip-isolation (Alperin et al. 1999). Macrofauna cell rupture during centrifugation was suggested to influence the extracted DOC, and the removal of macrofauna from sediments before centrifugation and whole-core squeezing was shown to reduce elevated DOC concentrations (Martin and McCorkle, 1993). In turn, our study site did not exhibit signatures of significant bioturbation (Dale et al., 2015). Herewith, at sites similar to our study area (low oxygen - low bioturbation), DOC concentrations extracted by centrifugation were in agreement either with those obtained by sip-isolation method (Komada et al., 2004) or with those obtained from in situ and ex situ incubations (Holcombe et al., 2001). Furthermore, Holcombe et al. (2001) suggested that sip-isolated pore-water DOC gradients may lead

to underestimation of diffusive DOC fluxes in low bioturbation regions. Thus, varying strength of organic matter–mineral associations may create different solute reservoirs around the surface of a mineral. Sip-isolation method was suggested to extract only loosely bound DOM out of the marine sediments, while centrifugation would sufficiently perturb sediments and sample the majority of the pore-water DOM that may efflux out of the sediment. In connection with the above, the centrifugation method was preferred as pore water extraction method for DOM analyses."

TK: Syringe filters can give large DOC background (and possibly DON also), but there is no mention about how the filters were cleaned. Please provide additional information showing that the data do not contain high (and variable) levels of blank. A: We thank the Dr. Komada for noticing that. Indeed, we did the mistake not to add the details behind choosing a filter type or their cleaning. Prior to the research cruise, we did several checks for different filters of that pore size, which are commonly used during pore water work, including PES, nylon, CA and RC. All the filters gave one or another background level, therefore, we tested which volume of ultrapure water was the optimal and reasonable for cleaning. CA and RC filters gave the minimal values for DOC and for DON after rinsing with 60 ml of ultrapure water among all filters. CA was chosen over RC due to lower binding affinity to macromolecules and proteins, as we did not want to influence recovery of organic components during filtration. Following will be added to the page 4 lines 25-30 to the revised version of the manuscript: "All samples were passed through pre-washed (60 mL of ultrapure water) cellulose acetate (CA) membrane syringe filters (0.2 $\mu$m). The preference for the CA filters was given as a result of a home-based test that occurred before the research cruise. Then, several types of filters (PES, nylon, CA and regenerated cellulose (RC)) were examined for background DOC and total dissolved nitrogen (TDN) signal. CA and RC filters gave the minimal background signal for both parameters after rinsing with 60 ml of ultrapure water (Fig. S4). CA was chosen over RC due to lower binding affinity to macromolecules and proteins"

Figure S4 will be also added to the Supplement.

TK: The authors state that microbial N turnover and DOM fluxes are likely related (page 9, line 11). I wholeheartedly agree with this statement, and find that this is an area that is ripe for further study. The authors go on to discuss N dynamics quite a bit, but the problem with this is that, other than DON, none of the inorganic N data are included in this manuscript. This renders most of the nitrogen-related discussion speculative at best. The authors should either scale back on the N discussion, include the DIN data, or perhaps plan on publishing a companion paper that includes relevant DIN data. At the very least, chamber data should include nitrate, assuming that was the major electron acceptor. The DIN data are also relevant to the extremely low DOC/DON ratios in sediments. The authors originally declare that nitrate/nitrite concentrations in sediments were negligible (bottom of page 5), then resurrect this issue as a possible explanation for the low DOC/DON ratio (bottom of page 9), only to dismiss it again (top of page 10). The authors provide a few other possible explanations for the low DOC/DON ratios, but this discussion would be a lot more convincing in the presence of a more complete DIN data showing that the DON values were not overestimated. A: Unfortunately, we were restricted by the data legacy and could not report on DIN from benthic chambers and pore waters, but could only use the data for our calculations of DON. The data on DIN from the benthic chambers will be published soon in a different manuscript by MSc David Clements and co-authors. However, MSc Clements has agreed to provide us the data for DIN for publishing from one of the stations. Therefore, data for DIN components from one benthic chamber at station 3 will be added to a Supplement as a Figure S6. Due to reviewers' suggestions, measurements of ammonia may now be published for all six stations and will be added to a Supplement as a depth profile plot (Figure S5).

TK: There seems to be an underlying assumption that sediment DOM is all refractory (e.g., page 1 line 5; page 2, line 21; page 11, line 15). As far as I am aware, this is not supported by the current literature. If anything, the opposite is more likely; a considerable fraction of DOM, especially near the sediment-water interface, is labile, and only a small fraction appears to be refractory (e.g., Bauer et al. 1995, Nature 373:686-689; Burdige et al. 2016, GCA 195:100-119; Komada et al. 2013, GCA 110:253-273). Therefore, assuming that the DOC and DON data presented here are indeed free of artifacts, it makes sense that the flux data point to microbial consumption of DOM that diffused out of the sediments. The authors should adjust the wording to better reflect the literature data. A:We thank Dr. Komada for her suggestion. Following will be added to the revised version of the manuscript on Page 2, line 33: "It was suggested previously that DOM in sediments consist of recalcitrant low molecular weight (LMW) compounds (Burdige and Gardner, 1998; Burdige and Komada, 2015), therefore, the sediment out-flux of DOM was hypothesized to serve an important source of recalcitrant DOM to the water column (e.g. Burdige and Komada, 2015; Burdige et al., 2016). Herewith, elevated concentrations of dissolved organic nitrogen (DON) within sediments suggest the presence of labile proteinaceous organic matter in pore waters, that have escaped degradation within the water column (e.g. Faganeli and Herndl, 1991). Furthermore, measurements and modelling of isotopic carbon composition in the anoxic and suboxic sediments off California, suggest that about 50 % of DOM within upper sediments represents isotopically young and labile DOM components, that are readily released to the water 5 column, where they are actively utilized by heterotrophs (Bauer et al., 1995; Komada et al., 2013; Burdige et al., 2016)."

Page 12, Line 8: "On the other hand, isotopic carbon composition suggests that a substantial fraction of pore-water DOM is isotopically young and is readily utilized by heterotrophic communities, when released to the water column (Bauer et al., 1995; Komada et al., 2013; Burdige et al., 2016)." Page 13 Line 3: "Thus, the production of humic-like LMWDOM along with the utilization of proteinaceous DOM suggest active microbial DOM utilization occurring in the near bottom waters (e.g. Alkhatib et al., 2013). Therefore, our results from the benthic chambers support the idea that DOM release to the water column may stimulate its utilization by water–column microbial communities (Komada et al., 2016; Burdige et al., 2016)." TECHNICAL COMMENTS:

TK: I had difficulty reading Fig. 3, because the panels are so small. I am also unable to tell the difference between dark grey and blue (DOC vs DON). A: DOC and DON will be separated in different panels in the revised version of the manuscript, we also will increase the size of the plot.

TK: Black and grey arrows in Fig. 9 also look identical in color. A: We will change the description under the plot to: "Conceptual view of DOM cycling near the sediment off Peru. Arrows directed out of the sediment represent diffusive fluxes of DOC (JDOC(Diff)) in mmolm-2d-1. Circular arrows indicate microbial DOM reworking, calculated as a difference of DOC (JDOC(Diff)) and net in situ flux DOC (JDOC(Net)) at each station."

TK: Written English is OK, but not in publishable shape (parts that would benefit from editing are too numerous to list here). A: We will address those issues with care.

TK: The narrative meanders in some places (e.g., discussion about DIN as I pointed out above). A: To avoid meandering we will omit following from the chapter 4.1: "For instance, NO3 that is present at high concentrations in intracellular vacuoles of Marthioploca (Dale et al., 2016) could be leaked to the pore water during sediment handling and centrifugation. An ammonium oxidizing bacteria were shown previously to be able profiting from nitrous oxide, produced by denitrification (e.g. Kartal et al., 2013). Thus, the production of NH+4, as a result of DNRA occurring at the inner shelf stations in combination to nitrous oxide production via denitrification occurring at outer shelf, may produce a convenient niche for anammox bacteria at the rim of the inner shelf at 12oS. The intermediate product of anammox, hydrazine (e.g. Kartal et al., 2013), may, in turn, accumulate in the inner space of anammox bacteria, and be released in the pore water samples as a consequence of the cell rapture induced by centrifugation. However, the concentrations of those intermediate products are likely very small and may not explain elevated TDN values."

TK: I also recommend streamlining the Introduction; I found the transition to DOC (line

16) a bit jarring. A: We will address this suggestion, e.g. following will be added to Page 2 line 16: "While POM degradation in sediments is mostly associated with its full remineralization to dissolved inorganic carbon (DIC) and inorganic nutrients, the mechanism of POM remineralisation implies important intermediate stages of dissolved organic matter (DOM) production, reworking and mineralization processes (Smith et al., 1992; Komada et al., 2013). Thus, around 10% of remineralized particulate organic carbon (POC) may accumulate as dissolved organic carbon (DOC) in the pore waters (Alperin et al., 1999). In turn, DOM efflux may represent an important escape mechanism for carbon from sediments (e.g. Ludwig et al., 1996; Burdige et al., 1999) and a source of organic matter to the water column (e.g. Burdige et al., 2016).Despite the acknowledged importance of sediment DOM for organic matter cycling, the measurements of benthic DOM fluxes remain scarce and the reactivity of the pore-water DOM is not well constrained."

---

## Author Comment (AC3) · 5 May 2020

We are very thankful to Dr. Piotr Kowalczuk for such a positive review! We will correct the typos in the revised version of the manuscript.

---

## Author Response (AR2)

**T.K.:** I'd like to thank the authors for making substantive improvements to the manuscript. Their responses to the comments I raised in the first review are thorough and thoughtful. This revised version is stronger than the first, but I still see issues with it that need to be addressed to get it into publishable shape.

**A.:** We are very thankful to Dr. Komada for taking her time and improving our manuscript!

**T.K.: Major comments:**

**T.K.:** First and foremost, the quality of writing is still poor. I made some edits, but stopped once I realized that there is a LOT to edit. Some sentences are barely comprehensible. I trust that with 8 authors, these can be identified and fixed. I list under Other Comments a small handful of issues that stuck out to me.

**A.:** We addressed the comments raised in "Other Comments". We hope that the manuscript has been improved now.

**T.K.:** Second, the data presentation and discussion seem overly complicated. I wonder if the narrative can be simplified to better clarify the take-home messages. Also, the authors toned down on the N story by a lot, but I still find that there is some overreach going on. Here's my argument:

The benthic chamber data can be interpreted in a more systematic way by comparing to the trend one would expect given the pore water data. For DOC and DON, we expect the values to increase with time. However, this is not always observed; some show no obvious change with time. Setting statistics aside for now, this opens the door for discussion about microbial DOM consumption at the sediment-water interface, or in the bottom water, for Sta. 3 and deeper. (Different matter for Sta. 1 and 2; see below re: Fig. 8.) For aCDOM(325), we also expect values to increase with time, but this is only seen at Sta. 4; elsewhere, values either remain unchanged, or decline. This again points to DOM consumption (or alteration) upon/during export from sediments. Similarly, trends in spectral slope S in the chambers are consistent with overall decline in MW of DOM with time; again pointing to DOM alteration upon/during export from sediments. And with FDOM, one finds that humic-like FDOM increased as expected, but perhaps not when it comes to protein-like FDOM, again pointing to DOM alteration/consumption. In its present state, these take-homes are difficult to grasp, because of too much attention being paid to subtle differences across time and stations (e.g., page 10 line18; p. 11 line 24).

One can then consider the observations that: (1) DON/DOC ratios were much lower in sediments compared to in the chambers, (2) and protein-like FDOM showed muted variability in the chambers, leading to the argument that proteinaceous DOM might have a role in this transformation (as authors discuss on page 11).

**A.:** We understand Dr. Komada's concern. We do try to address general patterns with those take-home messages mentioned by Dr. Komada in chapter 4.2. The middle shelf stations, however, rather did not fit general patterns. These differences could not be explained by the same processes, as for the rest of the stations. Therefore, we feel like the discussion on spatial variability is needed.

**T.K.:** I like the idea of Fig. 8, but after reading the manuscript twice, I am finding it a bit disingenuous. Perhaps I am missing something here, but it appears that this figure treats flux data from Sta. 1 and 2 rigorously (statistically speaking), and but not so for the rest of the stations. On page 9, line 14, the authors state that there were no significant differences between diffusive and net DOC fluxes. If that is the case, why is it that data from Sta. 3 – 6 are used to calculate DOM reworking rates, while this is not done for Sta. 1 and 2? In any case, reporting the reworking rates to two significant figures, and attempting to interpret them across a transect seems to be a stretch, given the level of uncertainty involved. We all know how difficult it is to get flux estimates, and that both diffusive and net fluxes are subject to systematic error. I wouldn't be surprised at all if there is indeed a bottom-water redox effect playing a role here, but whether the data clearly show that is a different matter.

**A.:** Despite the absence of statistical differences, which is mainly driven by the nonlinear patterns of our chamber values and, therefore, enormously large linear fit error, the apparent differences between flux estimates were very clear. They also followed similar pattern for stations on the outer shelf and the continental slope, therefore an apparent utilization was calculated for those stations. The middle shelf stations in turn generally exhibited very different distributions and patterns C:N, amino acid like fluorescence, etc. that made us hesitant in generalizing processes occurring there, therefore we added and would like to keep a separate discussion on the spatial variability of fluxes.

Herewith, we removed figure 8 as suggested by Dr. Komada.

**T.K.:** I find the discussion about pLMW-DOM on page 13 far-fetched. Unless the authors can provide a more concrete definition of pLMW-DOM and provide evidence for its existence in their samples, the discussion about unfolding of this material in presence of nitrate in the bottom waters should be omitted (paragraph starting at line 13). pLMW-DOM that is discussed in Burdige and Komada (2015) is largely theoretical, given very limited data on DOM size and composition.

**A.:** It is indeed theoretical. However, our FDOM and CDOM slope patterns in pore waters are controversial, while in the BIGO chambers they normally exhibited similar pattern. Thus, a humic-like FDOM increase suggests microbial DOM reworking, which is commonly accompanied by a decrease of molecular weight. The decrease in module of CDOM spectral slope suggests a relative increase in molecular weight. In our understanding, this could be resulting from polymerisation by hydrogen bonding, for instance. Then individual units would represent less bioavailable more complex and smaller molecules, but folding and electron density interactions could cause a decrease in the module of CDOM spectral slope, increasing general as we call it here "apparent molecular weight".

We rephrased our discussion: " The accumulation of humic-like fluorescence and its correlation with DOC concentrations (Comp.1, R=0.8, n=0.86, p<0.01), as observed during our study, has also been noted previously in marine sediments (e.g. Chen et al., 1993). The increase in the humic-like fluorescence with sediment depth is commonly explained as a net production of LMW recalcitrant humic DOM and an increasing fraction of FDOM in the porewaters compared to the water column (Komada et al., 2004). The increase in S275-295 over sediment depth indicated an increase in apparent molecular weight (Helms et al., 2008). This apparent increase in molecular weight in combination with the down-core enrichment in humic-like fluorescence may therefore suggest an accumulation of so-called polymeric LMW (pLMW) DOM. This may be formed via reactions of polymerisation (Hedges et al., 1988) or complexation (Finke et al., 2007), as well as due to formation

of supramolecular clusters via hydrogen bonding or hydrophobic interactions (e.g. Sutton and Sposito, 2005)."

**OTHER COMMENTS**

**TK:** The discussion about DOM removal either at the sediment-water interface, or in the bottom water, needs to be better clarified. The Abstract says sediment-water interface, but there seems to be some waffling going on in the narrative.

**A:** The "near-bottom water" and the "sediment-water interface" were used as synonymous here, as we may not distinguish between them by measurements in BIGO chambers. However, we understand the confusion and have edited our "waffling" on that matter. We have corrected that to "sediment-water interface" throughout the text.

**TK:** Also, is anything known about DOM in microbial mats? If they are covering ~40% of the surface, could they be playing a key role here? This point can be better clarified in the discussion.

**A:** We discuss the possibility of sediment release to stimulate the activity of microbial mats "Furthermore, DOM released by the sediment could potentially support an enhanced microbial abundance and carbon oxidation rates reported near the sediment on 12°S transect (Maßmig et al., 2020) and influence the activity of microbial mats that cover up to 100 % of the sediment surface at the middle shelf stations (Sommer et al., 2016)." Indeed, we believe, that especially in the middle shelf stations, bacterial mats may significantly affect the measured DOM fluxes and DOM quality. However, we refer to all kinds of heterotrophic microbial communities as potential DOM users as we may not distinguish the uptake quantitatively on the community level.

**T.K.:** I believe I mentioned this in my first review, but the sharp spikes in DOC and DON in the pore water profiles and in the chamber time series still bother me. Not because they exist, but that no plausible explanations are given for their occurrence.

**A:** Indeed, Dr. Komada addressed this question before. However, frankly, we may not explain those "spikes". We believe in our measured values, as not the sampling, nor the chemical analyses had an extra problem or measurement difficulty, and the samples were sampled, stored, and analysed in a similar way to all others. As well the peaks in porewaters occur for both FDOM and DOC at similar depth (although value-wise the differences are much smaller for FDOM than for DOC). If we would calculate the time corresponding to those peaks, using the sediment accumulation rates from Dale et al. (2015), the peaks would correspond to 4 and 49 years before the sampling. Checking ENSO variations did not give a general pattern, as 2013 corresponded to "normal year" after strong La Niña and 1966-1969 were characterized by strong El Niño with "normal" year in between.

**T.K.:** The net flux data are buried in Table S1. Upon looking at the numbers, many of the values look indistinguishable from zero. This point is not clear in the narrative (instead, tendency is to read too much into wiggles).

**A:** Please, note, that the values in the table S1 are not the net fluxes. Those values are changes in FDOM fluorescence and CDOM absorption calculated as unit/day. As those values do not represent concentrations in its classical understanding and may not accumulate (as Dr. Komada wisely corrected) we did not feel that it would be correct to account for the chamber volumes. Concerning the low values, we would like to refer to generally much lower FDOM and CDOM values in the water column compared to the pore waters. The increase in humic-like substances a day is comparable to the average FDOM QSE measured in the BIGO chambers. At the same time, the CDOM absorption in the chambers generally did not exhibit linear patterns, resulting in a very low rates and a big linear fit error.

**T.K.:** Please report the bottom water values along with t=0 values in the figure showing the chamber data.

**A:** Unfortunately, we did not have a bottom water sample for every station due to vial breakage during the transport. For our diffusive and net flux calculation, the t0 values from BIGO chambers were used. We did not use the bottom water values for those stations, where they were present, as we wanted to have all the same settings for flux calculations at each station.

"The initial concentrations in BIGO chambers and porewater solute concentrations from the uppermost slice of the sediment core (0 to 1 cm depth) were used for the flux calculations." Was added to the method section 2.4

**T.K.:** P1, L32: "…porewater DOM consists in part of recalcitrant low molecular weight…"

**A:** Corrected

**T.K:** P3, L1: Replace "Oh the other hand" with "At the same time"

**A:** "Oh the other hand" replaced  with "At the same time"

**T.K:** P3, L20-25: This section is very difficult to understand, and confuses the reader. Simplify, as you do on P12 L25, and clarify the direction in which absorbance drops.

**A:** We rephrased the sentence: "The S and absorption coefficients are used to learn on bulk DOM properties. For instance, a decrease of module value of S may indicate an increase in relative molecular weight (Helms et al, 2008)".

**T.K.:** P3, L25: I do not believe that all FDOM contain aromatic moieties.

**A:** We rephrased the sentence into: "The part of CDOM is fluorescent and it is mainly associated with aromatic molecular structures. This part of DOM is referred to as FDOM and is used to infer DOM quality (Coble 1996; Zlonay et al. 1999; Jorgensen et al., 2011; Catala et al., 2016; Loginova et al., 2016)."

**T.K.:** P3, L30-31: Example sentence that is barely comprehensible.

**A:** We deleted that sentence

**T.K.:** P4, L20-21: This comment about nitrate and nitrite is perplexing.

**A:** Changed into: "High concentrations of water column $NO_3^-$ and $NO_2^-$ were observed at stations deeper than 100 m water depth, while at shallower stations $NH_4^+$ was dominant dissolved inorganic nitrogen component. Thus, concentrations up to 1.2--1.4 µmol L-1 were detected in the middle shelf stations".

**T.K.:** P5: nice addition about syringe filters, including Fig. S4, thank you. But Fig. S4 is presented out of order (in fact, not all figures in the supplement are referenced)

**A:** Thank you for mentioning, were have changed the order and referenced all the plots.

**T.K.:** P5, L39: Multicorers (not multiple corers)

**A:** replaced

**T.K.:** P8, L10: Please explain how the range in D was incorporated when determining std deviations of fluxes.

**A:** We edited the part for calculating flux values: "The molecular size fractionation was not addressed during this study, therefore, we assumed that DOM molecular weight varied in the range from 0.5 to 10 kDa. This assumption resulted in D0 varying from 0.63x10^-6 to 7.2x10^-6 cm-2 s-1. This variance represented one of the major drivers of the estimated diffusive DOC (DON) flux variability. Therefore, the calculation of Js(Diff.) was done for the whole range of D0 with an increment of 0.1 x10^-6. Thus, Js(Diff.) presented in this manuscript is a resulting mean value of all the calculated Js(Diff.) and its variability expressed as a standard deviation".

**T.K.:** P8, L26-7: Eliminate the statement about vicinity to coast influencing DON distribution. (Or please re-write it, so that it does not sound as if DON concentration of a mud sample on a ship will change as you get closer to the coast.)

**A.:** the statement is eliminated

**T.K.:** P9, top line: what does it mean to "resume the gradient"? This appears again on line 25.

**A:** We remover the expression from all the sentences it was used

**T.K.:** P9, L7-8: What is meant by "near-bottom"? If you are referring to water enclosed in the chamber, say so. Here, DOC concentrations are for averages over time? Unclear.

**A:** Deleted. "Over time" added

**T.K.:** P9, L13-14: Instead of saying that net fluxes were 'generally lower' than diffusive fluxes, state that they were greater at Sta. 1 and 2. Provide plausible explanation. State whether diffusive and net fluxes differed for DON.

**A.:** The section was replaced with: "The diffusive DOC fluxes on the outer shelf and continental slope stations varied from a minimum of 0.2±0.1 mmol m-1 d-1 at St.4 to a maximum of 2.5±1.3 mmol m-1 d-1 at station 3 (St.3) (Fig. 4). Net in situ DOC fluxes determined with benthic chambers were lower than diffusive fluxes on those stations, varying from -0.3±0.9 at St.4 to 0.8±0.9 mmol m-1 d-1 at St.3. Net in situ DOC fluxes on the middle shelf stations were higher than fluxes estimated by Fick's law. Thus, the diffusive DOC fluxes were varying from 0.2±0.1 to 0.4±0.2 mmol m-1 d-1 and net in situ DOC were ranging between 1.1±0.9 and 2.3±2.3 mmol m-1 d-1. Diffusive DON fluxes ranged from -0.04±0.02 mmol m-1 d-1 at St.1 and St.6 to 3.3_1.7 mmol m-1 d-1 at St.2. Similar to DOC, net in situ DON fluxes were lower than diffusive DON fluxes on the outer shelf and continental slope stations, ranging from -0.05±0.3 mmol m-1 d-1 at St.6 to 0.3±0.3 mmol m-1 d-1 at St.5. In contrast to DOC fluxes, the diffusive DON flux on one of the middle shelf stations (St.2) was also higher than the net in situ DON flux, exhibiting 3.3±1.7 mmol m-1 d-1 and -0.03±0.3 mmol m-1 d-1, respectively. At St.1 both diffusive and net in situ DON flux estimates were very low. They displayed -0.04 ±0.02 mmol m-1 d-1 and 0.08±1.4 mmol m-1 d-1, respectively. Despite the clear apparent distinction between the different flux estimates for both, DOC and DON, no statistical differences were found between them at each station (p>0.05, Mann-Whitney Rank Sum Test, SigmaPlot, Systat Software)." In results. We are providing possible explanations for this matter in chapter 4.1.

**T.K.:** P9, L22: replace "absorption" with "absorption coefficient". I suggest "values of aCDOM(325)" instead of "aCDOM(325)s"

**A.:** "coefficient" has been added

**T.K.:** P9, L28: absorption coefficient can increase, but not accumulate. Sta.1 showed greater variance that what is suggested in text.

**A.:** Sorry, we did not notice a typo. It was corrected

**T.K.:** P9, L30: I would argue that the slope values were higher in the sediments compared to bottom water, but not necessarily increased with depth in sediment.

**A:** It did increase in the porewaters in most of the stations, especially comparing first 10 cm of the sediment cores

**T.K.:** P10, L12: Fluorescence can intensify, but not accumulate

**A:** This was corrected

**T.K.:** P10, L31: cite Fig. S1

**A:** Fig.S1. was cited

**T.K.:** P11: Top part of this page (and top part of second paragraph) is convoluted, and seems to delve too much into subtleties of the data that may not be significant. I find that it more obfuscating than clarifying.

**A.:** We thank Dr. T. Komada for this comment.

However, we would like to leave the discussion on spatial variability in the text. The middle shelf stations exhibit quiet different patterns for the sediment fluxes, compared to the outer shelf and continental slope stations and also compared to each other (for DON). These differences could not be explained by the same processes, as for the rest of the stations. Therefore, we feel like the discussion on spatial variability is needed. The outer shelf and continental slope stations exhibited somewhat similar patterns; therefore, the discussion does not treat the data so "rigorously".

**T.K.:** P12, to paragraph: Is there any methanogenesis going on here?

**A.:** Methanogenesis, according to the model by Dale et al., 2015 occurs at 60 cm of the sediment depth at st 1 and deeper at other stations.

**T.K.:** P12, 4.2 subheading: Please fix this.

**A.:** we hope that we fixed what Dr. Komada meant.

**T.K.:** P12, L10: replace "classical" with "current"; remove "slow" after "followed by"

**A.:** edited

**T.K.:** P12, L11: "can cause an imbalance in DOM"

**A.:** changed

**T.K.:** P12, L12: "This is in part explained by accumulation of recalcitrant DOM that is thought to be of LMW"

**A.:** changed

**T.K.:** P12, L14-16: sorption and coprecipitation would remove it from DOM

**A.:** Changed "contribute" to "affect"

**T.K.:** P12, L16-18: "isotope" needs to be identified.

**A.:** Rephrased to: "On the other hand, measurements of $\Delta^{14}C$ in the porewater DOM suggests that its substantial fraction is isotopically young and is readily utilised by heterotrophic communities, when released to the water column (Bauer et al., 1995; Komada et al., 2013; Burdige et al., 2016)."

**T.K.:** P13, L3: replace Fig. 8 with Fig. 4

**A.:** Changed

**T.K.:** P13, L5-6: "However, the previously reported … was generally not observed."

**A.:** edited

**T.K.:** P13, L10: I am not able to comprehend

**A.:** Changed into: "The decrease in S275-295 and enrichment in humic-like fluorescence over time indicated an accumulation of LMW humic DOM components (Helms et al., 2008). At the same time, the complex development of the amino acid-like fluorescence of Comp.3 and the drawdown of aCDOM(325) and DON, resulting in increased DOC/DON ratios, suggested a reworking of proteinaceous labile DOM (Fig.8). Therefore, the production of humic-like LMWDOM along with the utilisation of proteinaceous DOM suggests an active microbial DOM utilisation occurring at the sediment-water interface. These results support the idea that DOM release to the water column may stimulate respiration by water column microbial communities (Alkhatib et al., 2013; Komada et al., 2013; Burdige et al., 2016)."

**T.K.:** P13, L14: Komada et al (2016) should be replaced with (2013).

**A.:** replaced

**T.K.:** P13, L20-21: very hard to comprehend

**A.:** we rephrased to: "We suggest that the availability of electron acceptors, such as NO3- and NO2-, in the water column above the sediments could stimulate microbial communities at the sediment-water column interface to take up DOM".

**T.K.:** P14, L2-5: The wording is such that DOM data have built-in bias, while POM does not (as in good versus bad). I don't think the authors mean this; please re-word.

**A.:** Rephrased to: "In turn, POM respiration rates, which are commonly evaluated from DIC flux measured in benthic lander systems, may have been underestimated, as the diffusive DOC fluxes, calculated in this study could represent up to 53% of the estimated DIC flux (Clements et al., in prep.), and the net in situ benthic DOC fluxes could describe up to 28% of DIC flux.

**T.K.:** P14, L5 to end of paragraph: This can be shortened. I find it's obvious that bottom currents will move

**A.:** We appreciate that Dr Komada finds it obvious. However, Dr Komada is an experienced researcher, and often other (maybe less experienced authors) pay less attention to physical circulation addressing biogeochemical processes. We would like to keep this paragraph, as a

reminder for other authors of processes of physical circulation as one of the factors affecting biogeochemical processes.

**T.K.:** P14, second from bottom line: Should DOM be POM?

**A.:** Changed to "organic matter"

**T.K.:** P14, last row: the "previous studies" should be cited

**A.:** Citation has been added

**T.K.:** P15, L1: "decrease" is a poor choice of word here. "lower compared to"

**A.:** changed

**T.K.:** Fig. 1. "Left: Distribution of … stations (pentagrams)."

**A.:** edited

**T.K.:** Fig. 2. Alpha is used instead of "a" for abs coeff. Please use more distinct colors than black and dark blue. Panels for Sta. 1 should be expanded (or use line breaks) to show high data that aren't displayed. Also, as done in Fig. S1, will be helpful to indicate which stations are mid-shelf, etc.

**A.:** was changed

**T.K.:** Fig. 3. Were any of the data omitted from the regression?

**A.:** The t0 value and data included in brackets were excluded from the analyses. The brackets have been added to the plot.

**T.K.:** Fig. S1: Y-axis, NO4+ needs to be NH4+

**A.:** changed

**T.K.:** Fig. S3: What are dashed horizontal lines? Replace "near-bottom water" with "BIGO Chamber"

**A.:** Lines are deleted, indication changed.

**T.K.:** Fig. S4: Very informative, thank you.

**A.:** Our pleasure

**T.K.:** Were all figures in supplement referred to?

**A.:** Thank you for noticing! We referred now to all the supplement figures.

**T.K.:** increase/decrease "in", not "of"

**A.:** We edited this, thank you!

**T.K.:** "sediment-water interface", not "sediment-water column interface"

**A.:** we edited this, thank you!

**T.K.:** Intensity (such as fluorescence) can increase or decrease, but can't accumulate

**A.:** we edited this, thank you!

**T.K.:** Regions of the electromagnetic spectrum (UV, Vis) are 'ranges' or 'regions', not 'spectra'

**A.:** we edited this, thank you!

**Sediment release of dissolved organic matter to the oxygen minimum zone off Peru**

Alexandra N. Loginova[1], Andrew W. Dale[1], Frédéric A. C. Le Moigne[1,2], Sören Thomsen[1,3], Stefan Sommer[1], David Clemens[1], Klaus Wallmann[1], and Anja Engel[1]

[1]GEOMAR Helmholtz Centre for Ocean Research Kiel, Germany
[2]Mediterranean Institute of Oceanography, UM110, Aix Marselle Université, CNRS, IRD, 13288, Marselle, France
[3]LOCEAN-IPSL, IRD/CNRS/Sorbonnes Universites (UPMC)/MNHN, Paris, France

**Correspondence:** Anja Engel (aengel@geomar.de)

**Abstract.**

The eastern tropical South Pacific (ETSP) represents one of the most productive areas in the ocean that is characterised by a pronounced oxygen minimum zone (OMZ). Particulate organic matter (POM) that sinks out of the euphotic zone is supplied to the anoxic sediments and utilised by microbial communities. The degradation of POM is associated with the production and

5   reworking of dissolved organic matter (DOM). The release of DOM to the overlying waters may represent an important organic matter escape mechanism from remineralisation within sediments but received little attention in OMZ regions so far. Here, we combine measurements of dissolved organic carbon (DOC) and dissolved organic nitrogen (DON) with DOM optical properties in the form of chromophoric (CDOM) and fluorescent (FDOM) DOM from porewaters and near-bottom waters of the ETSP off Peru. We evaluate diffusion-driven fluxes and net *in situ* fluxes of DOC and DON to investigate processes affecting DOM

10   cycling at the sediment-water interface along a transect at $12^{\circ}$S. To our knowledge, these are the first data for sediment release of DON and porewater CDOM and FDOM for the ETSP off Peru. Porewater DOC accumulated with increasing sediment depth, suggesting an imbalance between DOM production and remineralisation within sediments. High DON accumulation resulted in very low porewater DOC/DON ratios ($\leq 1$) which could be caused by an "uncoupling" in DOC and DON remineralisation. Diffusion driven fluxes of DOC and DON exhibited high spatial variability and ranged from $0.2\pm0.1$ mmol m$^{-2}$d$^{-1}$ to $2.5\pm1.3$

15   mmol m$^{-2}$d$^{-1}$ and from $-0.04\pm0.02$ mmol m$^{-2}$d$^{-1}$ to $3.3\pm1.7$ 
[revised manuscript text omitted]